# Optimizing Network Simulation: Enhancing Performance Prediction Accuracy via Neural Architecture Search

**ShaoChen He** [1]   **Zirui Zhuang** [1]   **Haifeng Sun** [1]   **Xiaoyuan Fu** [1]   **Qi Qi** [1]   **Lei Zhang** [2]   **Jianxin Liao** [1]   **Jingyu Wang** [1]

## Abstract

Existing machine learning models for network simulation excel at predicting average performance but, due to their reliance on mean squared error, systematically fail to capture the critical tail latency and jitter that define modern network stability. This "tail-blindness" renders them unreliable for latency-sensitive systems. We bridge this gap by introducing Accurate Neural Architecture Search (ANAS), a paradigm that automates the discovery of architectures for high-precision, distribution-aware network simulation. ANAS corrects the evaluation inaccuracies of weight-sharing NAS via a similarity-constrained search, employs a hybrid search space to model complex traffic, and uses a Wasserstein loss to optimize for the entire delay distribution, not just its mean. Empirically, the ANAS-discovered architecture is holistically superior: it reduces overall validation loss by 25.8% compared to DeepQueueNet, demonstrating strong average-case performance, while simultaneously excelling at tail-sensitive metrics by lowering the normalized Wasserstein distance ($W_n$) by up to 69.8%. This confirms its ability to faithfully model a comprehensive performance spectrum, encompassing both average and critical tail behaviors. The ANAS framework provides a practical methodology for automatically creating high-fidelity models of network devices, enabling more reliable validation of next-generation network protocols and algorithms.

[1]State Key Laboratory of Networking and Switching Technology, Beijing University of Posts and Telecommunications, Beijing, China [2]China Unicom, Beijing, China. Correspondence to: Zirui Zhuang <zhuangzirui@bupt.edu.cn>.

*Proceedings of the 43rd International Conference on Machine Learning*, Seoul, South Korea. PMLR 306, 2026. Copyright 2026 by the author(s).

## 1. Introduction

For many of today's latency-sensitive applications, performance is no longer judged solely by averages. Metrics such as throughput and mean delay remain indispensable for assessing baseline efficiency. Yet, in distributed AI training, cloud gaming, and video conferencing, system success or failure increasingly hinges on the tail of the distribution: the 99th or even 99.9th percentile of delay and jitter. A single microsecond-scale fluctuation can stall a multimillion-dollar GPU cluster (Dean & Barroso, 2013; Li et al., 2019; Lao et al., 2021; Sapio et al., 2021). While averages remain relevant, this shift demands a new class of simulation tools capable of capturing the entire performance distribution—both the body and its critical tail.

Traditional packet-level simulators (Riley & Henderson, 2010; Varga, 2010) excel at capacity planning by preserving fine-grained visibility into queuing and scheduling. However, their reliance on sequential event processing makes them prohibitively slow at modern scales; even parallel variants (PDDES) (Jafer et al., 2013; Riley & Henderson, 2010) are often bottlenecked by synchronization overheads, rendering them impractical for rapid, iterative design. Conversely, machine learning-based estimators like RouteNet (Xie et al., 2018b; Rusek et al., 2020), M3 (Li et al., 2024), and Simnet (Li et al., 2022) achieve high efficiency by abstracting traffic into flows or paths. While this design facilitates large-scale analysis of average trends, it sacrifices packet-level visibility: the microscopic dynamics driving jitter, burst-induced congestion, and synchronization failures are no longer directly represented. Ultimately, current approaches expose a fundamental trade-off: traditional tools offer visibility but lack scalability, while ML abstractions offer efficiency but lack the fidelity required for tail-sensitive workloads.

A milestone in this direction was DeepQueueNet (Yang et al., 2022), which demonstrated that a general-purpose neural model could capture device-level behavior with packet-level visibility. Despite its impact, subsequent studies (Fu et al., 2021; Wu et al., 2022) and our experiments show that its prediction errors are disproportionately concentrated in jitter and tail latency. The root cause is twofold. First, the standard Mean Squared Error (MSE) objective prioritizes

fitting the bulk of the distribution, effectively treating rare tail events as noise. Second, and more critically, its fixed, general-purpose backbone lacks the structural inductive bias required to capture extreme network volatility. Such rigid architectures tend to over-smooth the microscopic bursts and non-linear interactions that drive worst-case performance.

Attempts to mitigate this via manual architecture crafting or extensive fine-tuning are costly and unscalable. Manual design relies on expensive trial-and-error, while fine-tuning demands vast data yet often fails to generalize. More fundamentally, human expertise faces significant challenges in manually identifying the optimal structural complexity required to capture elusive tail behaviors. To fundamentally resolve this, we must move beyond fixed, generic backbones. What is needed is a systematic, automated paradigm capable of **tailoring model architectures to specific network scenarios**, ensuring the structural complexity aligns with the intrinsic demands of the traffic distribution.

These shortcomings motivate **Neural Architecture Search (NAS)** as a systematic alternative to automate design exploration. However, directly applying mainstream weight-sharing NAS (e.g., ENAS (Pham et al., 2018), DARTS (Liu et al., 2018)) to network simulation exposes a fundamental mismatch. While NAS has excelled in classification tasks that tolerate relative logit inaccuracies (Koonce, 2021; Tan & Le, 2019), packet-level simulation is inherently a high-precision regression problem. Within this context, slight inaccuracies in delay prediction can amplify into qualitative errors in tail statistics. As a result, the evaluation noise introduced by weight-sharing—typically benign in classification—becomes detrimental to regression, rendering architecture ranking unreliable and the search ineffective (Yu et al., 2020). Thus, the central question arises: **How can NAS be tailored to meet the stringent high-precision demands of network simulation regression?**

To answer this, we propose **Accurate Neural Architecture Search (ANAS)**, a paradigm tailored to the rigorous demands of network simulation. Our approach is grounded in the insight that weight-sharing bias necessitates a similarity-aware evaluation protocol: evaluations are trustworthy only when candidates are structurally similar to the currently optimized path, ensuring ranking consistency. Concretely, ANAS employs a **similarity-constrained evaluation** strategy that filters noisy scores to restore reliable architecture ranking. Structurally, we support this search with a **hybrid attention–recurrent space**, which unifies long-range flow dependencies with local queueing dynamics. Finally, to ensure the discovered architectures capture extreme volatility, we integrate a **distribution-aware Wasserstein objective**. This explicitly aligns optimization with the full delay distribution, preserving the efficiency of NAS while elevating it to meet the precision required for reliable digital twins.

In summary, our contributions are:

- Similarity-constrained search: We leverage similarity with dynamic warm-up to improve evaluation reliability under weight sharing, ensuring accurate and efficient architecture ranking.

- Attention–recurrent search space: We design a seq2seq architecture that unifies long-range dependency modeling with fine-grained queue/scheduling dynamics, outperforming conventional baselines.

- Distribution-aware training: By incorporating the Wasserstein distance, ANAS learns the entire delay distribution, leading to significant gains in both average-case fidelity and tail-sensitive metrics.

Empirically, ANAS identifies an optimal architecture within 16 hours, reducing validation loss by 25.8% compared to DeepQueueNet. Across static traffic models, ANAS reduces normalized Wasserstein distance $(W_n)$ by up to 98.8% relative to RouteNet, 69.8% relative to DeepQueueNet, and 43.9% relative to ENAS. In dynamic scenarios, it lowers $W_n$ by 67.8% versus DeepQueueNet and 44.8% versus ENAS. These results confirm that ANAS closes the long-standing gap between efficiency and fidelity, offering a robust simulation paradigm that captures the holistic performance distribution, effectively spanning from averages to long-tail latencies.

## 2. Related Works

### 2.1. Network Simulation Paradigms

Network simulation typically navigates a trade-off between fidelity and efficiency. **Packet-level discrete-event simulators** (e.g., ns-3 (Riley & Henderson, 2010), OMNeT++ (Varga, 2010)) provide granular visibility but incur prohibitive runtime costs, as even parallel execution faces synchronization bottlenecks (Riley & Henderson, 2010). While **stream-level simulators** offer some efficiency gains (Robertazzi, 2000; Bolch et al., 2006), they lack fine-grained inspection capabilities. To bypass these limits, **end-to-end estimators** utilize neural networks to directly predict performance (Rusek et al., 2020; Li et al., 2022; 2024). However, these "black-box" methods often face scrutiny regarding interpretability and scalability limitations (Fu et al., 2021).

Bridging this gap, **DeepQueueNet** (Yang et al., 2022) introduces a two-stage paradigm (Fig. 1): it first trains a packet-level model to capture device behavior and then instantiates it across a target topology. This approach preserves observability while enhancing speed. However, its efficacy relies heavily on a manually specified architecture. As highlighted in (Ferriol-Galmés et al., 2022), manual designs often fail

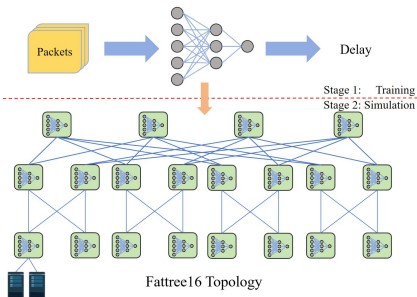

*Figure 1.* Schematic of the generic two-stage simulation process.

to capture complex queuing dynamics, where suboptimal architectural choices significantly impair accuracy.

## 2.2. NAS and Ranking Challenges in Regression

**Neural Architecture Search (NAS)** automates model design. Early agent-based methods (RL, evolutionary algorithms) (Zoph & Le, 2016; Jaafra et al., 2019) explore vast spaces but are computationally expensive. To improve efficiency, **weight-sharing NAS** evaluates candidates via a supernet (Pham et al., 2018; Bender et al., 2018; Feng et al., 2024) or jointly optimizes parameters (Liu et al., 2018; Ye et al., 2022; Xie et al., 2018a). Despite their speed, these methods face a critical challenge: inherited weights often misalign with fully trained outcomes, leading to performance collapse where the search favors overly simplistic architectures (Chen et al., 2021; Zhang et al., 2020). Systematic studies confirm a weak correlation between supernet validation and true performance (Bender et al., 2018; Yu et al., 2020). In **high-precision regression tasks** like network simulation, such evaluation noise undermines the optimization of stringent tail objectives, necessitating a robust, regression-aware evaluation protocol.

## 3. Methodology

### 3.1. Overall Framework

Accurate Neural Architecture Search is an automated framework designed to discover task-specific neural architectures optimized for network simulation. As shown in Fig. 2, ANAS operates in three phases: Dynamic Supernet Warm-up to stabilize shared weights; Similarity-Guided Search to filter noisy evaluations using our inter-architecture similarity metric; and Architecture Derivation, which uses a performance predictor and LSTM agent to derive the final topology.

Formally, ANAS addresses a device-level architecture search problem. Given training data, validation data, and a search space $\mathcal{S}$, the goal is to identify an architecture $G^* \in \mathcal{S}$ that achieves the lowest validation loss after being

trained from scratch with the hybrid objective defined later in this section. Because fully training every candidate architecture is computationally infeasible, ANAS relies on two search-time proxies: shared-weight supernet evaluation to collect architecture–performance signals, and a controller reward derived from the validation loss. These proxies are used only to guide the search; the final deployed predictor is the selected architecture retrained from scratch rather than the controller or the supernet itself.

### 3.2. Core Components of ANAS

Before detailing the search process, we introduce the two fundamental components of our framework: the search space that defines the universe of possible architectures, and the controller that explores this space.

#### 3.2.1. THE HYBRID SEARCH SPACE

As illustrated in Fig. 3, our hybrid search space is defined over a full encoder–decoder architecture designed for sequence-to-sequence tasks. The encoder is a Directed Acyclic Graph (DAG) consisting of a fixed input node (node 0) and $n$ intermediate computational nodes (e.g., nodes 1–3). A second DAG forms the decoder, which contains $m$ intermediate nodes (e.g., nodes 5–7) and uses the encoder's final state as its own initial hidden state. In parallel, a multi-head self-attention mechanism (Vaswani et al., 2017) processes the encoder's output to generate a weighted context vector. This context is then concatenated with the original input to provide a richer signal to the decoder, enhancing its ability to focus on relevant input features.

During the search, the NAS agent's task is to determine the optimal connections between the intermediate nodes within each DAG and to select an activation function (e.g., ReLU, sigmoid, tanh, or swish) for each node. Each intermediate node is a recurrent neural network (RNN) cell augmented with a highway connection (Zilly et al., 2017), which uses a learned gate $c$ to balance feature transformation and information preservation:

$$y = c \otimes f(x \cdot W^h) + (1-c) \otimes x, \quad c = \text{sigmoid}(x \cdot W^c). \quad (1)$$

This flexible, component-based design creates an expansive search space. For instance, with $n = 6$ intermediate nodes in the encoder and $m = 6$ in the decoder, the number of possible architectures reaches approximately $1.5 \times 10^{10}$, enabling the discovery of highly expressive, task-specific models.

#### 3.2.2. THE LSTM CONTROLLER

An LSTM-based controller, or agent, is responsible for navigating the vast search space by learning a policy to generate promising architectures. The agent dynamically samples an architecture by sequentially selecting the input connections

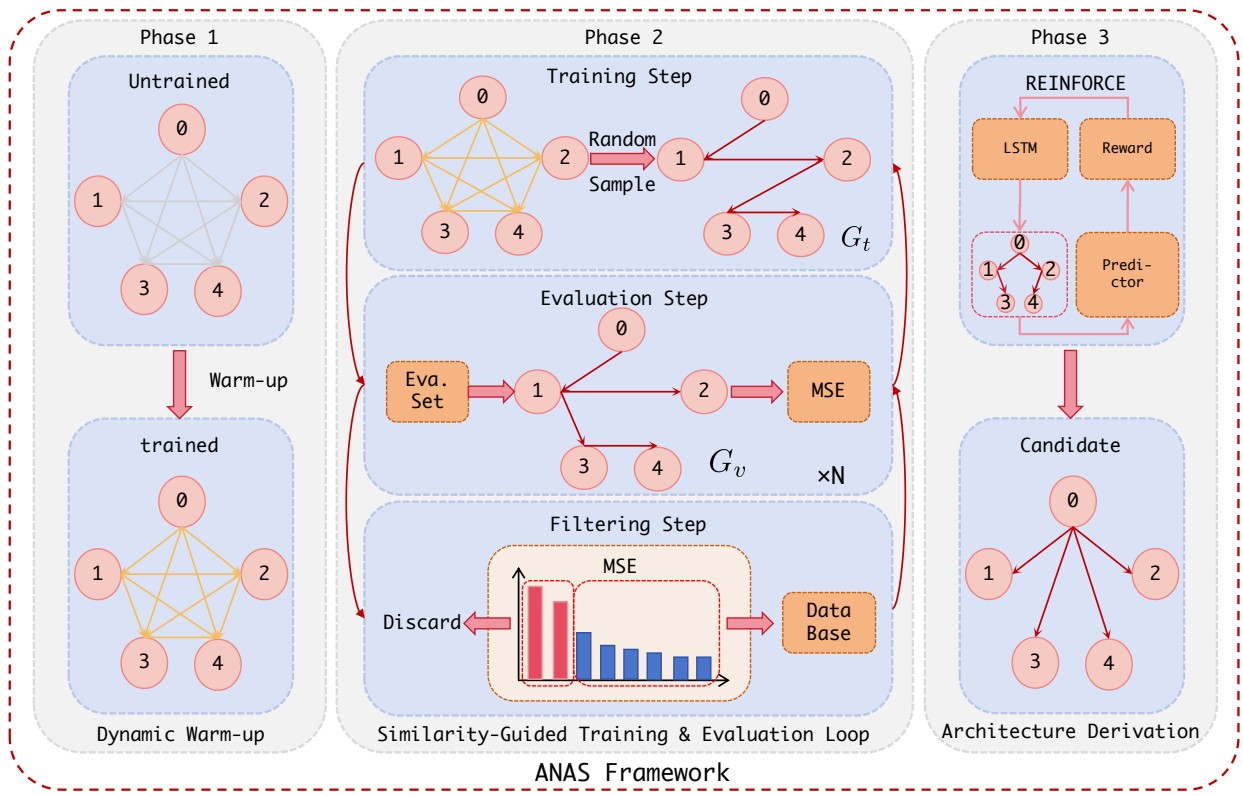

*Figure 2.* The three-phase framework of the ANAS algorithm.

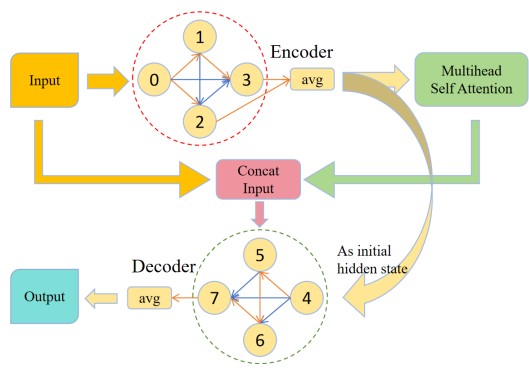

*Figure 3.* Model architecture and hybrid search space schematic.

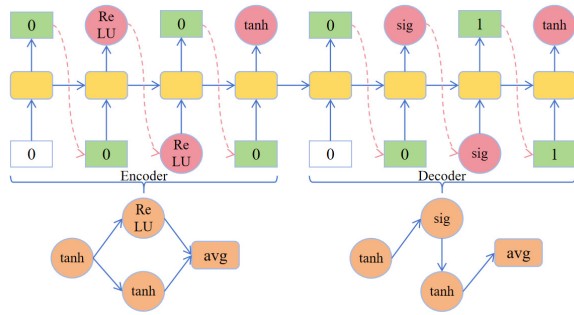

*Figure 4.* An example of the agent sampling an architecture with $n = m = 3$ nodes.

and activation functions for each node within the encoder and decoder DAGs. Fig. 4 illustrates this process for an example with $n = m = 3$ nodes, showing how the unrolled LSTM controller makes a series of discrete decisions to construct the final DAG structures. Once an architecture is fully defined, its performance (i.e., validation loss) is either fetched from collected data or estimated by the performance predictor during the architecture derivation phase. This performance score is then converted into a reward signal. The agent's policy is updated using the REINFORCE algorithm (Williams, 1992) with a moving baseline, allowing it to progressively favor sampling architectures that are predicted to perform well.

### 3.3. The ANAS Search Process in Detail

#### 3.3.1. PHASE 1: DYNAMIC SUPERNET WARM-UP

The objective of dynamic warm-up is to mitigate evaluation inaccuracies caused by insufficient weight training in the early stages. This phase ensures that the shared weights within the supernet are adequately stabilized before being used for architecture evaluation. During this phase, ANAS uniformly samples architectures and trains each for a pre-defined number of batches. This process is repeated until a

*Table 1.* Effect of Inter-architecture Similarity on L1 Loss ($\times 10^{-3}$)

| $S_A$ | **Mean** | **Median** | **Max** |
|-------|----------|------------|---------|
| 0.958 | 9.6 | 8.8 | 20.1 |
| 0.917 | 10.8 | 9.8 | 25.2 |
| 0.875 | 11.6 | 10.6 | 27.3 |
| 0.833 | 12.3 | 11.3 | 29.6 |
| 0.792 | 12.9 | 11.8 | 31.3 |
| 0.750 | 13.2 | 12.1 | 33.1 |
| 0.701 | 13.7 | 12.6 | 32.3 |
| 0.667 | 13.7 | 12.9 | 33.8 |

termination criterion is met (e.g., when the supernet's loss converges below a threshold), enhancing the reliability of subsequent evaluations.

### 3.3.2. PHASE 2: SIMILARITY-GUIDED TRAINING AND EVALUATION

In this phase, ANAS focuses on acquiring high-fidelity architecture–performance data. Standard weight-sharing NAS (e.g., ENAS (Pham et al., 2018)) typically alternates between a supernet Training Step and a controller Evaluation Step. However, this paradigm presents an inherent limitation: the supernet weights become transiently optimized for the specific structural properties of the most recently updated architecture. Consequently, evaluating a structurally dissimilar candidate using these biased weights yields a performance estimate that reflects the candidate's proximity to the previous path rather than its intrinsic potential. This weight-coupling noise degrades the reliability of the reward signal, hindering the controller's ability to accurately rank architectures.

Our key insight is that this evaluation bias is strongly correlated with the structural similarity between the training architecture and the evaluated architecture. To address this, we introduce inter-architecture similarity, $S_A$, which measures the structural resemblance by comparing the proportion of matching edges ($E$) and activation functions ($F$) between a training architecture ($G_t$) and a candidate architecture ($G_v$):

$$S_A = \frac{|(E_t \cap E_v) \cup (F_t \cap F_v)|}{|E_t| + |F_t|}. \tag{2}$$

As shown in Table 1, our experiments confirm this insight: as similarity ($S_A$) to the training architecture decreases, the evaluation error (L1 loss) and its variance progressively increase. This demonstrates that evaluating architectures dissimilar to the one used for training yields unreliable, noisy performance signals.

Based on this finding, we replace the standard evaluation process with a more robust three-step loop designed to collect high-fidelity performance data. Each iteration proceeds as follows:

1. **Training Step:** A single training architecture, $G_t$, is randomly sampled, and the supernet's shared weights are updated by training this architecture for one step.

2. **Evaluation Step:** Immediately following this, a batch of candidate architectures, $\{G_v\}$, is generated by applying a minimal number (1–2) of perturbations to $G_t$. This *High-Similarity Evaluation* strategy ensures all candidates are evaluated in a region where the just-optimized supernet weights can provide reliable performance estimates.

3. **Filtering Step:** After the candidates are evaluated on the validation set, the top 10% of results with the highest loss are identified and discarded as outliers, ensuring that only high-confidence data is retained.

The remaining set of reliable (architecture, performance) pairs generated from this loop is then used to update the controller's policy in Phase 3. This ensures the controller learns from a low-noise signal, leading to a more stable and effective search over time. A direct ranking-correlation analysis based on independently retrained architectures is provided in Appendix A.5, further confirming that similarity-guided evaluation yields a meaningful search signal.

### 3.3.3. PHASE 3: ARCHITECTURE DERIVATION

Using the reliable set of architecture–performance pairs collected in the previous phase, ANAS efficiently identifies the optimal architecture. To do this, we train a performance predictor using XGBoost (Chen & Guestrin, 2016). We select this gradient-boosted tree model because it fits tabular architecture representations effectively and maintains high prediction fidelity even with limited training samples, outperforming other surrogate methods in our experiments. During search, the predictor serves as a surrogate scorer: when the controller proposes an architecture for which no ground-truth evaluation has been collected, the predictor estimates its validation loss so that controller training can continue without incurring the cost of repeated true evaluations.

The controller is trained using the REINFORCE algorithm (Williams, 1992). To align the RL objective of maximizing rewards with the NAS goal of minimizing validation loss, the reward function $R$ is defined as being inversely proportional to the predicted validation loss $L_{\mathrm{mse}}$:

$$R(\cdot) = \frac{c}{L_{\mathrm{mse}}}, \quad c = 5 \times 10^{-4}. \tag{3}$$

This inverse relationship ensures that the agent is rewarded for discovering architectures with lower loss. The scaling constant $c$ modulates the magnitude of the reward signal. It ensures the reward is scaled to a numerically stable range suitable for policy-gradient updates, preventing both overly

weak signals (which slow convergence) and overly strong signals (which cause instability).

After the controller's training converges, we sample ten candidate architectures. These candidates undergo short-term training and validation comparison, and the architecture yielding the lowest validation loss is selected as the final candidate for full retraining from scratch. In this way, the predictor improves search efficiency and sampling quality, but the final architecture is determined by actual validation performance rather than predictor output alone.

### 3.4. Distribution-Aware Loss Function

A key challenge in network simulation is that performance metrics like packet delay often follow long-tailed distributions, where rare, high-delay events (the "tail") are critical performance indicators. Traditional loss functions such as Mean Squared Error (MSE) are ill-suited for such data. MSE minimizes the average squared difference across all data points. In a long-tailed distribution, the model can achieve a low overall MSE by accurately predicting the vast majority of common, low-delay packets in the distribution's body, even if it performs poorly on the few but critical packets in the tail. This biases the model toward the average case at the expense of the worst case, systematically failing to capture the very phenomena that network engineers often care about most.

To address this, ANAS employs a hybrid loss function that combines MSE with the Wasserstein distance ($L_w$). The Wasserstein term acts as a regularization mechanism that prevents overfitting to the bulk of the distribution by explicitly penalizing deviations in the tail. While MSE anchors the prediction to the ground truth on average, the Wasserstein distance quantifies the difference between the predicted and actual probability distributions, making it inherently sensitive to discrepancies in the tail. For 1D distributions with Cumulative Distribution Functions (CDFs) $U$ and $V$, the Wasserstein loss is:

$$L_w(p, q) = \int_{-\infty}^{+\infty} |U(x) - V(x)| \, dx. \tag{4}$$

The total loss is a weighted sum: $L(\cdot) = \alpha \cdot L_{\text{mse}} + \beta \cdot L_w$. We set $\alpha = 1$ and $\beta = 0.01$ based on extensive sensitivity analysis (see Appendix D). The analysis confirms that this value achieves the optimal trade-off: it improves final MSE by approximately 5% compared to pure MSE ($\beta = 0$) while maintaining training stability, whereas larger values introduce oscillations that degrade convergence quality.

## 4. Experiment

### 4.1. Experimental Setup

#### 4.1.1. TASK AND DATASETS

The primary task is to train a sequence-to-sequence model that accurately predicts the behavior of packets passing through a 4-port switch. We use the public dataset provided with DeepQueueNet (Yang et al., 2022) for training and validation. This dataset was generated using a packet-level simulator (ns.py) with a Markovian Arrival Process (MAP) traffic generator. It comprises 3500 packet flows, each spanning 16041 time steps, resulting in 16000 data points per flow for sequence-to-sequence learning. The packet scheduler in the switch supports multiple queuing disciplines, including first-in-first-out (FIFO), strict priority (SP), deficit round robin (DRR), and weighted fair queuing (WFQ). For SP, packet priorities are randomly assigned values between 1 and 3. For DRR and WFQ, weights are randomly selected from 1 to 9. The load factor for each port is maintained between [0.1, 0.8] by adjusting the strengths of these mechanisms.

#### 4.1.2. BASELINES FOR COMPARISON

We benchmark the architecture discovered by ANAS against four distinct categories of baselines to evaluate different aspects of performance. First, to compare against existing domain standards, we select **DeepQueueNet (DQN)**, the current state-of-the-art manually designed model, and **ENAS**, a representative weight-sharing NAS algorithm. Second, we include powerful general-purpose sequence models, **LSTM** and **Transformer**, to determine whether domain-specific architecture specialization is truly necessary compared to off-the-shelf solutions. Finally, we establish a rigorous control using **Random Search** allocated with **twice the computational budget** of ANAS. This comparison serves to strictly validate that our performance gains from efficient search strategy rather than computational expenditure.

For the end-to-end network simulation task, we select **RouteNet** and **DQN** as they represent the established state-of-the-art methods in this domain, and **ENAS** as the representative SOTA for architecture search. Detailed comparisons with general-purpose architectures (LSTM, Transformer) are provided in the Appendix A.4 to maintain focus on domain-specific benchmarks.

#### 4.1.3. EVALUATION METRICS

Model accuracy is evaluated using two primary metrics. **Mean Squared Error (MSE)** measures the average prediction error. The **normalized Wasserstein distance ($W_n$)** is more sensitive to distributional differences between the predicted and ground truth delays, especially in the tail. It

is defined as:

$$W_n(\hat{y}, y) = \frac{W(\hat{y}, y)}{W(\mathbf{0}, y)}, \tag{5}$$

where $\hat{y}$ denotes the predicted values, $y$ represents the ground truth, and $W(\cdot, \cdot)$ is the Wasserstein distance. For both metrics, lower values indicate better performance.

### 4.1.4. IMPLEMENTATION DETAILS

**ANAS Search Configuration** The architecture search process begins with a dynamic warm-up phase, where resampling occurs every 400 batches and terminates after 30 samples if no new optimal loss is observed over five consecutive resamplings. In the subsequent training and evaluation phase, resampling is performed every 800 batches, with each sampling evaluating 1000 architectures. The perturbation probabilities for generating candidate architectures are: 25% for perturbing the encoder twice, 25% for perturbing the decoder twice, and 50% for perturbing each once. The LSTM agent has a hidden dimension of 200 and is trained with the Adam optimizer (Kingma & Ba, 2014) at a learning rate of 0.00035 for 20000 steps. To encourage exploration, we use a tanh scaling constant of 2.5, a temperature of 5.0, and add the agent's sample entropy to the reward with a weight of 0.0001. The supernet is trained with the Adam optimizer at a learning rate of 0.001.

**Final Model Training Configuration** The final architecture discovered by ANAS, as well as all baseline models, are trained from scratch for a fair comparison. For our model, the encoder and decoder use hidden layer dimensions of 140, and the attention layer is set to 128 dimensions, divided into four heads of 32 dimensions each. The model is trained to minimize the hybrid loss function using the Adam optimizer with an initial learning rate of 0.002, which decays by 0.92 each epoch. Training is performed for 20 epochs with a batch size of 512. The entire process, including search and final training, requires approximately 48 hours on a single Nvidia Tesla P100-12GB GPU.

**Baseline Settings and Controlled Variables** To ensure fairness, we align the parameter count of all models (including Transformer/LSTM) to the SOTA DeepQueueNet ($\approx$664k). We categorize the training configurations to isolate performance sources:

- **Standard Baselines** (DQN, ENAS, LSTM, Transformer): Trained with the conventional **MSE loss** to benchmark existing methods in their canonical, unoptimized configurations.

- **Controlled Baselines** (ANAS, Random Search): Trained using our **Hybrid Loss** within the proposed **Hybrid Search Space**. This setup establishes Random Search as a rigorous control: by sharing the identical

*Table 2.* Final model performance of different methods. Note that MSE values are scaled by $10^{-5}$, and $W_n$ metrics are scaled by $10^{-3}$.

| Method | Params | MSE | $\mathbf{W_n}$ | $\mathbf{p99W_n}$ | GPU[a] |
|---|---|---|---|---|---|
| **DeepQueueNet** | 664K | 6.93 | 10.7 | 21.7 | - |
| **LSTM** | 664K | 8.43 | 19.0 | 27.8 | - |
| **Transformer** | 664K | 5.96 | 6.0 | 22.1 | - |
| **ENAS** | 667K | 6.10 | 11.9 | 20.0 | 22 h |
| **Random Search** | 657K | 5.64 | 6.1 | 18.8 | 32 h |
| **ANAS** | **657**K | **5.14** | **4.5** | **17.0** | **16** h |

[a] Refers to the GPU time used to search architecture.

search space and loss function, we isolate the search strategy as the sole variable.

### 4.2. Traffic Modeling Performance

Table 2 compares ANAS against baselines on the held-out test set. ANAS achieves state-of-the-art performance, outperforming all competitors in MSE, $W_n$, and importantly, the extreme tail metric $p99W_n$.

**Search Efficiency and Component Validation** Crucially, ANAS outperforms the Double-Budget Random Search in half the time (16h vs. 32h). Since both share the same search space and loss, this strictly isolates the efficiency of our strategy. Furthermore, Random Search (using our Hybrid components) achieves a lower $p99W_n$ (18.8) than ENAS (20.0), validating that our **Hybrid Search Space and Loss** provide a superior foundation for tail-sensitive modeling compared to standard setups.

**Comparison with Manual and General Baselines** ANAS significantly surpasses the manual SOTA (DQN), reducing MSE by **25.8%**. The contrast is even more revealing against the general-purpose Transformer: while the Transformer achieves a competitive MSE, it fails to capture the extreme tail, yielding a high $p99W_n$ of 22.1 ($10^{-3}$). In contrast, ANAS reduces this metric to 17.0, demonstrating that only our specialized architecture effectively suppresses errors in the worst-case scenarios (99th percentile), which is critical for robust network simulation. To isolate the effect of the training objective, we additionally report fixed-backbone results under the same hybrid loss in Appendix A.4. Those controlled results confirm that the gains cannot be attributed to the loss alone.

### 4.3. End-to-End Simulation and Generalization

To evaluate the downstream impact, we deploy the ANAS-discovered architecture as a reusable digital twin within end-to-end network simulations. We assess its accuracy and robustness across three distinct categories of network conditions. First, we utilize **Static Scenarios** (MAP, Poisson, and On–Off processes) to validate simulation fidelity

*Table 3.* Path-wise $W_n$ comparison across static (\*), dynamic (†), and real-world (‡) scenarios. Metrics cover average and 99th percentile (p99) delay and jitter. DQN denotes DeepQueueNet; YTL refers to the YouTube Live trace.

| Scenario | Method | avgDelay | p99Delay | avgJitter | p99Jitter |
|---|---|---|---|---|---|
| MAP\* | RouteNet | 0.044 | 0.014 | 0.041 | 0.030 |
| | DQN | 0.017 | 0.039 | 0.031 | 0.035 |
| | ENAS | 0.011 | 0.010 | 0.015 | **0.012** |
| | **ANAS** | **0.005** | **0.010** | 0.013 | 0.015 |
| Poisson\* | RouteNet | 0.674 | 0.972 | 1.951 | 1.083 |
| | DQN | 0.009 | 0.014 | 0.055 | 0.035 |
| | ENAS | 0.004 | 0.016 | 0.010 | **0.008** |
| | **ANAS** | **0.001** | **0.012** | **0.005** | 0.009 |
| On–Off\* | RouteNet | 0.549 | 0.578 | 1.420 | 0.847 |
| | DQN | 0.006 | 0.051 | 0.016 | 0.022 |
| | ENAS | 0.010 | 0.037 | 0.025 | 0.022 |
| | **ANAS** | **0.002** | **0.020** | **0.004** | **0.005** |
| BT† | DQN | 0.011 | 0.031 | 0.048 | 0.048 |
| | ENAS | 0.010 | 0.026 | 0.026 | 0.014 |
| | **ANAS** | **0.003** | **0.015** | **0.008** | **0.004** |
| HL† | DQN | 0.034 | 0.085 | 0.067 | 0.081 |
| | ENAS | 0.036 | 0.078 | 0.036 | 0.046 |
| | **ANAS** | **0.026** | **0.059** | **0.010** | **0.023** |
| YTL‡ | DQN | **0.017** | 0.022 | 0.146 | 0.060 |
| | ENAS | 0.175 | 0.289 | 0.389 | 0.278 |
| | **ANAS** | 0.020 | **0.006** | **0.009** | **0.020** |
| Roblox‡ | DQN | 0.063 | 0.110 | 0.287 | 0.109 |
| | ENAS | 0.338 | 0.857 | 0.856 | 0.314 |
| | **ANAS** | **0.031** | **0.087** | **0.124** | **0.031** |

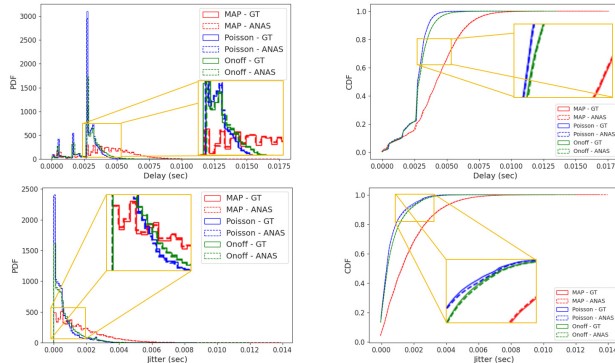

*Figure 5.* Delay and Jitter Distribution Comparison Between ANAS and Ground Truth under the static MAP traffic model.

under steady-state loads. Second, we introduce **Dynamic Scenarios**, specifically High Load (HL) and Burst Traffic (BT), to test model robustness against operational volatility such as congestion and link failures. Finally, to capture complex intricacies not modeled by synthetic generators, we employ **Real-World Application Traces** (YouTube Live (YTL) and Roblox) from a recent 5G corpus[1]. These traces are characterized by sharp bursts of activity followed by extended periods of silence, presenting the most rigorous test for the model's long-tail modeling capabilities. Detailed scenario configurations, packet dataset statistics, and extensive evaluations on additional scenarios are provided in Appendix A.1.

**Performance in Static Scenarios**    Table 3 details the simulation fidelity across different scenarios. In the static scenarios (\*), ANAS demonstrates a comprehensive advantage. Averaged across all scenarios, ANAS reduces the error metrics by **98.8%** compared to RouteNet, **69.8%** compared to DQN, and **43.9%** compared to ENAS. Figure 5 visually corroborates this fidelity, plotting the delay and jitter distributions under the MAP model. The near-perfect overlap between ANAS's predictions (dotted lines) and the ground truth (solid lines) establishes it as an excellent tool for accomplishing routine tasks such as protocol verification.

**Robustness in Dynamic Scenarios**    Dynamic scenarios introduce transient congestion and volatility, causing inevitable performance degradation across all models com-

---

[1]The dataset is available at https://www.kaggle.com/datasets/kimdaegyeom/5g-traffic-datasets/data.

pared to static baselines. However, ANAS demonstrates superior **resilience**. In the High Load scenario, ANAS effectively mitigates the sharp degradation seen in standard baselines. It maintains an average delay of 0.026, significantly outperforming the 0.034 recorded by DQN. This robustness is further maximized in the Burst Traffic scenario, where ANAS leverages its hybrid loss to capture rapid state changes. Consequently, it reduces *p99Jitter* to **0.004**, representing a 70% improvement over ENAS. These results confirm that ANAS provides the most reliable "digital twin" capability for complex network environments, maintaining consistency even where absolute accuracy is challenged.

**Generalization to Real-World Traces**    Real-world traces like *YouTube Live* and *Roblox* pose severe challenges due to their distinct "on-off" patterns. Unlike synthetic data, these workloads consist of extended silence interspersed with sharp bursts, forcing predictions almost exclusively into the **long-tail region**. Under this distribution shift, **ENAS** fails to generalize. On the *Roblox* trace, it yields a p99Delay of 0.857, which is nearly ten times worse than the 0.087 achieved by ANAS. This failure is likely due to architecture specialization on synthetic training data. Conversely, ANAS utilizes its flexible Hybrid Search Space and tail-aware Wasserstein Loss to capture these unseen extreme dynamics. The advantage is evident in *YouTube Live*: despite a marginally higher average delay than DQN, ANAS delivers a **93% reduction in average jitter** and a **66% reduction in p99 jitter**. This confirms that ANAS successfully prioritizes **distributional stability**, ensuring reliability under the most volatile conditions.

Finally, a detailed analysis of simulation efficiency is provided in Appendix B, confirming that ANAS achieves its state-of-the-art accuracy without sacrificing the computational efficiency advantages of two-stage simulation methods.

*Table 4.* Ablation study of ANAS components. Performance is measured by MSE ($\times 10^{-5}$), $W_n$ ($\times 10^{-3}$), and $p99W_n$ ($\times 10^{-3}$) on the test set. The baseline is Random Search with a double budget.

| Configuration | MSE | $W_n$ | $p99W_n$ |
|---|---|---|---|
| ANAS (Full Model) | 5.14 | 4.5 | 17.0 |
| w/o Wasserstein Loss | 5.32 | 4.8 | 18.1 |
| w/o Similarity & Warm-up | 5.78 | 5.7 | 19.1 |
| w/o Hybrid Search Space | 5.94 | 6.3 | 19.8 |
| Baseline: Random Search (32h) | 5.64 | 6.1 | 18.8 |

### 4.4. Ablation Study

To quantify component-wise contributions, we conduct an ablation study by systematically disabling core innovations from the full ANAS model. Table 4 summarizes the resulting degradation in MSE, $W_n$, and the tail-critical $p99W_n$, using the double-budget Random Search as a strict performance baseline.

**Impact of the Hybrid Search Space**  The Hybrid Search Space is the most critical factor. Restricting the search to a conventional cell-based space causes the severest degradation, increasing MSE by **15.6%** to 5.94 and $W_n$ to 6.3. Notably, this performance trails even the Random Search baseline, indicating that the search strategy alone cannot overcome the representation bottleneck of a constrained space. The global context modeling inherent in the full encoder-decoder architecture is therefore fundamental for superior performance.

**Impact of Similarity-Guided Evaluation**  The inter-architecture similarity metric and dynamic warm-up phase are essential for search efficiency. Removing these mechanisms raises the MSE by **12.4%** to 5.78. This degradation confirms that the dynamic warm-up stabilizes the supernet weights while the similarity metric filters evaluation noise, preventing the controller from being misled by inaccurate signals. Without them, the effective exploration of the search space is severely compromised.

**Impact of the Wasserstein Loss**  The Wasserstein loss functions as a critical refinement for distributional accuracy. While disabling it leads to a modest **3.5%** increase in MSE, the impact on tail metrics is more pronounced. The $W_n$ increases to 4.8, and the tail-sensitive $p99W_n$ degrades from 17.0 to 18.1. These shifts confirm that the Wasserstein loss effectively forces the model to capture rare, high-delay events that standard MSE-based optimization tends to overlook, ensuring better alignment with the true traffic distribution.

In summary, the ablation study confirms that the hybrid search space provides the foundational capacity, similarity-guided mechanisms ensure efficient exploration, and the Wasserstein loss fine-tunes the model for tail distribution accuracy. Collectively, these components enable ANAS to achieve state-of-the-art performance.

## 5. Conclusion

In this paper, we tackle the **evaluation inaccuracy** of weight-sharing NAS in high-precision regression tasks. We propose **ANAS**, a framework that employs a **similarity-guided evaluation** strategy to rectify supernet bias and ensure reliable architecture ranking. ANAS discovers architectures that achieve state-of-the-art performance in modeling complex traffic dynamics, particularly regarding **jitter and tail latency**. This advancement enables the creation of trustworthy "digital twins" for network infrastructure, facilitating precise capacity planning and bottleneck diagnosis.

Beyond specific performance gains, ANAS provides a robust methodology for applying AutoML to complex engineering systems. By shifting the focus from simply finding architectures to ensuring the **trustworthiness of the search process**, ANAS bridges the gap between the promise of NAS and the rigorous demands of network simulation, paving the way for more intelligent and resilient infrastructure.

## Acknowledgements

This work was supported in part by the National Key R&D Program of China under Grant 2024YFE0200800; in part by the National Natural Science Foundation of China (62471055, U23B2001, 62401080, 62406039, 62321001, 62201072, 62101064); in part by the Fundamental and Interdisciplinary Disciplines Breakthrough Plan of the Ministry of Education of China (JYB2025XDXM107); in part by the High-Quality Development Project of the Ministry of the MIIT (2440STCZB2584); in part by the Fundamental Research Funds for the Central Universities; in part by the Ministry of Education and China Mobile Joint Fund (MCM20200202, MCM20180101); in part by the 2025 Education and Teaching Reform Project Funding at Beijing University of Posts and Telecommunications (2025YZ005).

## Impact Statement

This paper advances machine learning for high-fidelity network simulation. We do not identify any societal consequence requiring specific discussion here.

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

*Table 5.* Overview of datasets and scenarios used in our experiments (LF: Link Failures). Synthetic rows report per-run settings; real-trace rows report properties of the original captures.

| Dataset | Duration | #Packets | Burst | LF |
|---------|----------|----------|-------|-----|
| *Synthetic Datasets* | | | | |
| MAP | 30 s | 288,192 | | |
| Poisson | 30 s | 288,192 | | |
| On–Off | 30 s | 288,192 | | |
| LB | 30 s | 368,337 | ✓ | ✓ |
| RI | 30 s | 362,089 | ✓ | ✓ |
| BT | 30 s | 364,762 | ✓ | ✓ |
| HL | 30 s | 441,173 | ✓ | ✓ |
| *Real-world Traces* | | | | |
| Anarchy | ∼1.1 h | 964,842 | ✓ | |
| YouTube Live | ∼3.6 h | 914,413 | ✓ | |
| GeForce NOW | ∼1.1 h | 3,957,337 | ✓ | |
| Netflix | ∼24.7 h | 4,290,778 | ✓ | |
| Zoom | ∼5.8 h | 6,095,138 | ✓ | |
| Roblox | ∼25.1 h | 928,379 | ✓ | |

## A. Supplemental Experimental Details

To validate the robustness of ANAS across diverse network conditions, we employ a comprehensive suite of datasets ranging from controlled synthetic baselines to complex real-world application traces. Table 5 provides a statistical overview of all scenarios used in our experiments.

The datasets are categorized into two distinct groups:

- **Synthetic Datasets:** These include both static baselines (MAP, Poisson, On-Off) and dynamic stress tests (LB, RI, BT, HL). As shown in the table, these scenarios are configured with a fixed duration of 30 seconds, where packet counts denote the average traffic volume per simulation run.

- **Real-World Traces :** These represent production traffic patterns captured from live environments. For these rows, the table reports the duration and total packet counts of the original captures, which serve as the ground truth for our trace-driven simulations.

Detailed configurations for the dynamic scenarios and the provenance of the real-world traces are provided in the following subsections.

### A.1. Dynamic Scenario Configurations

As summarized in the synthetic section of Table 5, the four dynamic scenarios (HL, LB, BT, RI) are designed to stress robustness under network volatility. All dynamic scenarios share two characteristics: (i) bursty traffic patterns, where the arrival rate alternates between normal and high-intensity phases; and (ii) random link failures, which transiently disrupt connectivity.

- **High Load (HL).** High-utilization conditions with frequent short-lived congestion episodes triggered by intense bursts along a primary path. The per-port load factor varies from **0** to **0.9**. Each run contains on average **441,173** packets.

- **Load Balancing (LB).** Steady baseline traffic with Poisson arrivals ($\lambda$=15) while periodically injecting burst phases and random link failures. The per-port load factor varies from **0.1** to **0.5**. Each run contains on average **368,337** packets.

- **Burst Traffic (BT).** Non-stationarity introduced by selected flows transitioning into burst mode at intervals, creating sharp load spikes. The per-port load factor varies from **0** to **0.7**. Each run contains on average **364,762** packets.

- **Random Intensity (RI).** Moderate irregularity by sampling the Poisson rate from a time-varying process (Gaussian-perturbed intensity), together with bursts and link failures. The per-port load factor varies from **0.1** to **0.5**. Each run contains on average **362,089** packets.

### A.2. Real-Trace Datasets

The real-world evaluation utilizes traces from two sources, as detailed in the bottom section of Table 5:

**Anarchy Online (server-side).** A server-side MMORPG packet trace from a commercial provider (Funcom), published at MMSys 2012. This dataset is representative of interactive gaming traffic captured directly at the game server.

**5G mobile application subsets.** Subsets of a 328-hour 5G corpus collected in South Korea (May–Oct 2022) using PCAPdroid on a Samsung Galaxy A90 5G. Captures are per-application without background traffic, covering OTT VoD (*Netflix*), live streaming (*YouTube Live*), video conferencing (*Zoom*), metaverse/gaming (*Roblox*), and cloud gaming (*GeForce NOW*). Traces are timestamped packet-header time series suitable for per-application traffic modeling.

### A.3. Extended Performance on Complex Scenarios

To rigorously evaluate the model's generalization capabilities under realistic and highly volatile conditions, we conducted extensive tests using the real-world application traces and additional dynamic scenarios described in Section A.1.

Table 6 presents the quantitative comparison between ANAS and the state-of-the-art baselines. The results reveal con-

sistent performance characteristics regarding the trade-off between latency minimization and distributional stability:

1. **Prioritizing Stability in Real-Time Applications:** For interactive applications like *GeForce Now* (cloud gaming) and *Anarchy* (MMO), the consistency of packet arrival (jitter) is often more critical than the raw average speed. In the *GeForce Now* scenario, ANAS demonstrates a distinct optimization preference: while it yields a slightly higher average delay than ENAS (0.044 vs. 0.025), it achieves a superior reduction in tail instability. Specifically, ANAS lowers the *p99Jitter* to **0.002**, an order of magnitude lower than both ENAS (0.031) and DQN (0.015). This indicates that the Wasserstein loss successfully guides the model to suppress micro-bursts and maintain a smooth delay distribution, rather than merely fitting the mean at the expense of variance.

2. **Robustness in High-Bandwidth Streaming:** In the *Netflix* scenario, characterized by heavy buffering bursts, ANAS demonstrates comprehensive dominance. It achieves the lowest errors across all four metrics, reducing the p99Delay to 0.067 compared to 0.082 for ENAS. This confirms that for high-throughput traffic, the architecture can simultaneously optimize for both latency and stability without compromise.

3. **Resilience to Structural Volatility:** The synthetic dynamic scenarios (LB and RI) introduce structural volatility through random link failures and route changes. Under these conditions, baseline methods often degrade due to their inability to adapt to topology shifts. In contrast, ANAS maintains performance levels comparable to static scenarios, achieving extremely low jitter (avgJitter $\approx$ 0.005) compared to DQN ($\approx$ 0.058). This suggests that the discovered architecture effectively captures the underlying physics of queuing dynamics, rendering it robust to external topological variations.

### A.4. Architecture Comparisons Beyond the Main Baselines

**General-purpose architectures.** To assess whether off-the-shelf sequence models are sufficient for network simulation, we compare ANAS with general-purpose architectures under the same parameter budget.

We benchmark ANAS against two powerful general-purpose baselines: **Transformer** and **LSTM**. To ensure a fair comparison regarding model capacity, all models were constrained to a parameter budget similar to the SOTA DeepQueueNet ($\approx$664k).

*Table 6.* Performance comparison on real-world application traces and synthetic dynamic scenarios.

| Dataset | Method | avgDelay | p99Delay | avgJitter | p99Jitter |
|---|---|---|---|---|---|
| *Real-world Application Traces* | | | | | |
| Anarchy | DQN | 0.015 | 0.061 | 0.073 | 0.082 |
| | ENAS | 0.017 | 0.038 | 0.049 | 0.040 |
| | **ANAS** | 0.015 | **0.013** | **0.033** | **0.027** |
| GeForce Now | DQN | 0.059 | 0.012 | 0.008 | 0.015 |
| | ENAS | **0.025** | 0.031 | 0.031 | 0.031 |
| | **ANAS** | 0.044 | **0.009** | **0.007** | **0.002** |
| Zoom | DQN | 0.173 | 0.150 | 0.193 | 0.108 |
| | ENAS | 0.036 | **0.027** | 0.014 | 0.050 |
| | **ANAS** | **0.022** | 0.050 | **0.009** | **0.031** |
| Netflix | DQN | 0.058 | 0.117 | 0.115 | 0.136 |
| | ENAS | 0.037 | 0.082 | 0.042 | 0.088 |
| | **ANAS** | **0.016** | **0.067** | **0.036** | **0.079** |
| *Synthetic Dynamic Scenarios* | | | | | |
| LB | DQN | 0.018 | 0.046 | 0.058 | 0.061 |
| | ENAS | 0.017 | 0.036 | 0.018 | 0.015 |
| | **ANAS** | **0.012** | **0.029** | **0.005** | **0.004** |
| RI | DQN | 0.018 | 0.048 | 0.057 | 0.060 |
| | ENAS | 0.016 | 0.035 | 0.026 | 0.014 |
| | **ANAS** | **0.012** | **0.029** | **0.005** | **0.004** |

*Table 7.* Path-wise $W_n$ of different methods across scenarios for average delay, jitter, and 99th percentile delay, jitter.

| Scenario | Method | avgDelay | p99Delay | avgJitter | p99Jitter |
|---|---|---|---|---|---|
| MAP | Transformer | 0.01 | 0.019 | 0.018 | 0.020 |
| | LSTM | 0.015 | 0.013 | 0.017 | 0.030 |
| | **ANAS** | **0.005** | **0.010** | **0.013** | **0.015** |
| Poisson | Transformer | 0.006 | 0.016 | 0.021 | 0.01 |
| | LSTM | 0.013 | 0.038 | 0.018 | 0.009 |
| | **ANAS** | **0.001** | **0.012** | **0.005** | 0.009 |
| On–Off | Transformer | 0.01 | 0.033 | 0.03 | 0.016 |
| | LSTM | 0.022 | 0.067 | 0.04 | 0.028 |
| | **ANAS** | **0.002** | **0.020** | **0.004** | **0.005** |

Table 7 details the path-wise normalized Wasserstein distance ($W_n$) on static datasets. The results demonstrate that:

- While Transformer and LSTM models can achieve reasonable accuracy on *average* delay metrics, they struggle to capture the *tail* distribution and jitter.

- ANAS consistently outperforms the general-purpose models across all scenarios. Notably, in the *Poisson* scenario, ANAS reduces the p99Delay error by approximately 25% compared to the Transformer, validating that the hybrid search space and domain-aware loss function are critical for modeling the extreme tail behaviors inherent in network queues.

**Controlled backbone comparison under the hybrid loss.** To separate the effect of the training objective from that of the architecture, we additionally retrained **DeepQueueNet**, **Transformer**, and **LSTM** under the same MSE + Wasserstein objective used by ANAS, while keeping the matched-parameter setting unchanged.

*Table 8.* Controlled backbone comparison under the same hybrid loss. MSE values are scaled by $10^{-5}$, and $W_n$ metrics are scaled by $10^{-3}$.

| Method | MSE | $W_n$ | $p99W_n$ |
|---|---|---|---|
| DeepQueueNet | 6.48 | 5.33 | 18.15 |
| Transformer | 6.19 | 6.54 | 18.53 |
| LSTM | 8.26 | 12.32 | 25.01 |
| **ANAS (Ours)** | **5.14** | **4.5** | **17.0** |

*Table 9.* Comparison with alternative similarity metrics. Mean/median MSE values are scaled by $10^{-4}$, and max loss is scaled by $10^{-3}$.

| Metric | Spearman | Mean | Median | Max |
|---|---|---|---|---|
| Simple | -0.389 | 3.93 | 3.14 | 1.76 |
| Depth-aware | -0.414 | 3.92 | 3.12 | 1.79 |
| Path-aware | -0.400 | 3.91 | 3.10 | 1.81 |
| Combined | -0.394 | 3.92 | 3.10 | 1.82 |

Table 8 shows that sharing the same training objective does not eliminate the gap to the final searched architecture. Even under the hybrid loss, all fixed backbones remain worse than ANAS on the main evaluation metrics, indicating that the performance gains cannot be explained by the loss design alone. At the same time, the controlled results reveal clear structural trade-offs among the fixed backbones: **Transformer** is relatively stronger on average error (MSE), **DeepQueueNet** is relatively more competitive on tail-sensitive distribution metrics, and **LSTM** remains the weakest overall. This pattern suggests that ANAS benefits from the combination of a distribution-aware objective and a searched architecture whose structural inductive bias is better aligned with tail-sensitive network simulation.

### A.5. Additional Analyses of Similarity-Guided Evaluation

**Ranking correlation and discovered architecture patterns.** We further analyze whether similarity-guided supernet evaluation preserves a meaningful architecture ranking. From 17,613 candidate architectures, we uniformly sampled 30 models across supernet ranking percentiles, retrained each from scratch, and compared its supernet validation MSE with its fully trained best validation MSE.

The resulting rank-correlation statistics are **Kendall tau = 0.370** and **Spearman = 0.527**. These positive coefficients show that similarity-guided evaluation provides a meaningful ranking signal: architectures ranked more favorably by the supernet also tend to achieve lower validation loss after independent retraining.

As a qualitative reference, Yu et al. (Yu et al., 2020) reported an average Kendall tau of approximately $-0.004$ for standard weight-sharing NAS in RNN search spaces. Although the tasks and search spaces are not identical, this comparison suggests that ANAS substantially mitigates ranking distortion under weight sharing rather than merely improving an internal search proxy.

Beyond ranking consistency, the search logs also reveal recurring structural preferences among high-performing architectures. Compared with the bottom 10% of sampled architectures, the top 10% exhibit an approximately 12% smaller maximum decoder depth (2.65 vs. 3.00), about 15%

more output branches (3.10 vs. 2.70), and roughly 18% lower average validation loss when terminal nodes connect back to earlier states through skip connections. These patterns suggest that ANAS does not merely identify isolated high-performing samples; rather, the learned search signal consistently favors structural biases that are better aligned with tail-sensitive network simulation.

**Alternative similarity metrics.** We next tested whether more complex structural similarity definitions would materially improve the reliability signal used in Phase 2. In addition to the original **Simple** metric used in the paper, we considered **Depth-aware**, **Path-aware**, and **Combined** variants. All results were computed on the evaluated candidates after the 10% filtering step.

Table 9 shows that richer metrics can provide slightly finer-grained discrimination. For example, the depth-aware variant achieves a marginally stronger Spearman correlation than the simple metric. However, the gains are small: the mean and median MSE values remain very close across all variants, while the simple metric attains the lowest maximum loss. Therefore, the adopted simple metric is not an arbitrary simplification; it already captures the main reliability signal while preserving low complexity and interpretability.

**Sensitivity to the filtering threshold.** Finally, we examine whether the top-10% filtering ratio is overly sensitive. Under the same similarity setting and with the other hyperparameters unchanged, we computed the retained statistics after trimming the highest-loss samples at different ratios from the same batch of local candidate evaluations.

Table 10 shows two key patterns. First, the untrimmed 0% setting is clearly worse: compared with 0%, 10% trimming reduces the mean MSE from $2.81 \times 10^{-4}$ to $2.51 \times 10^{-4}$ and the maximum loss from $1.20 \times 10^{-3}$ to $3.79 \times 10^{-4}$. This supports our interpretation that local shared-weight evaluation contains a small but influential extreme-noise tail. Second, the method is not sensitive around the chosen threshold: increasing the trim ratio from 10% to 15% or 20% changes the mean and median only slightly. Therefore, the role of 10% filtering is not to act as a finely tuned optimum, but to provide a robust trade-off between removing extreme noisy samples and retaining enough data for subsequent

*Table 10.* Sensitivity analysis of the filtering threshold in Phase 2 local evaluation. All loss values are scaled by $10^{-4}$.

| Trim ratio | Mean | Median | Max |
|---|---|---|---|
| 0% | 2.81 | 2.43 | 12.0 |
| 5% | 2.60 | 2.41 | 5.04 |
| 10% | 2.51 | 2.40 | 3.79 |
| 15% | 2.45 | 2.39 | 3.22 |
| 20% | 2.41 | 2.38 | 2.85 |

*Table 11.* Parallel inference time for a 30-second simulation.

| Method | GPUs used | Time |
|---|---|---|
| OMNeT++ | 0 (CPU) | 2h 22m 11s |
| MimicNet | 1 | 4m 02s |
| DeepQueueNet | 1 | 5m 12s |
| | 2 | 2m 45s |
| | 4 | 1m 27s |
| ANAS | 1 | 5m 24s |
| | 2 | 3m 04s |
| | 4 | 1m 40s |

predictor and controller training.

## B. Simulation Efficiency Analysis

We verify that ANAS's accuracy does not compromise computational performance by measuring inference time for a 30-second simulation on a fattree16 topology. Results in Table 11 show that ANAS's runtime is comparable to other two-stage methods (e.g., DeepQueueNet) and orders of magnitude faster than traditional discrete-event simulators.

## C. Supplementary Algorithms for ANAS

This section provides the detailed pseudocode for the ANAS algorithm described in Section 3. Unlike the high-level overview in the main text, the following algorithms explicitly present the training loops, candidate perturbation, filtering steps, and reinforcement learning updates that constitute the complete search procedure.

### C.1. Shared Weights Training and Reliable Data Collection (Phase 1 & 2)

Algorithm 1 corresponds to Sections 3.3.1 and 3.3.2. It describes (i) the *Dynamic Warm-up Phase*, where shared weights in the supernet are stabilized via uniform sampling, and (ii) the *Similarity-Guided Training and Evaluation Phase*, where candidate architectures are generated by perturbing the training architecture, evaluated on the validation set, and filtered to retain reliable architecture-performance pairs.

**Algorithm 1** Shared Weights Training and Reliable Data Collection (Phases 1 and 2)

**Require:** Supernet $\mathcal{M}$, training set $\mathcal{D}_{train}$, validation set $\mathcal{D}_{val}$, warm-up budget $B_w$, collection budget $B_c$, candidate size $N = 800$
**Ensure:** Reliable architecture-performance dataset $\mathcal{D}_{arc}$
1: **Warm-up Phase:**
2: **for** $i = 1$ **to** $B_w$ **do**
3:     Sample architecture $G_t$ uniformly from search space $\mathcal{S}$
4:     Train $\mathcal{M}(G_t)$ on minibatch from $\mathcal{D}_{train}$
5:     Update shared weights by backpropagation
6: **end for**
7: **Data Collection Phase:**
8: **for** $j = 1$ **to** $B_c$ **do**
9:     Sample training architecture $G_t$ from $\mathcal{S}$
10:     Train $\mathcal{M}(G_t)$ for one step on $\mathcal{D}_{train}$
11:     Generate candidates $\{G_v\}_{i=1}^{N}$ by perturbing $G_t$ (edges or activations)
12:     **for all** $G_v$ in $\{G_v\}$ **do**
13:         Evaluate validation loss $L_{val}(G_v)$ on $\mathcal{D}_{val}$
14:         Store $(G_v, L_{val}(G_v))$ in buffer
15:     **end for**
16:     Discard top 10% candidates with highest loss
17:     Append remaining pairs to $\mathcal{D}_{arc}$
18: **end for**
19: **return** $\mathcal{D}_{arc}$

### C.2. Controller Training and Architecture Derivation (Phase 3)

Algorithm 2 corresponds to Section 3.3.3. Using the reliable dataset collected in Phase 2, an LSTM-based controller is trained with REINFORCE, where rewards are inversely proportional to validation loss (Eq. 4 in the main text). After training, the controller samples multiple candidate architectures, each retrained from scratch, and the best-performing one on the validation set is returned as the final architecture.

## D. Wasserstein Loss Weight Sensitivity Analysis

### D.1. Motivation

The hybrid loss function $L(\cdot) = \alpha \cdot L_{\text{mse}} + \beta \cdot L_w$ introduces a critical hyperparameter $\beta$. While the MSE term ensures point-wise accuracy, it is prone to overfitting—focusing on local noise rather than the global distribution. The Wasserstein term ($L_w$) is introduced specifically to counteract this by enforcing distributional consistency. This section analyzes how $\beta$ acts as a regularizer to prevent overfitting.

**Algorithm 2** Controller Training and Architecture Derivation (Phase 3)

**Require:** Controller policy $\pi_\theta$, predictor $\mathcal{P}$, reliable dataset $\mathcal{D}_{arc}$, training data $\mathcal{D}_{train}$, validation data $\mathcal{D}_{val}$, reward constant $c = 5 \times 10^{-4}$, baseline $b$ (sliding mean), derive steps $D$

**Ensure:** Final architecture $G^*$

1: **Controller Training:**
2: **for** $t = 1$ **to** $T$ **do**
3:      Sample architecture $G_t \sim \pi_\theta$
4:      Predict validation loss $\hat{L} = \mathcal{P}(G_t)$
5:      **if** $\hat{L}$ is missing **or** below threshold **then**
6:          Train $\mathcal{M}(G_t)$ and evaluate true $L_{val}(G_t)$
7:      **end if**
8:      Compute reward $R = \frac{c}{L_{val}(G_t)}$ {Eq. (4)}
9:      Update controller parameters $\theta$ via REINFORCE using baseline $b$
10: **end for**
11: **Architecture Derivation:**
12: Sample $D$ candidate architectures $\{G_d\}$ from controller
13: **for all** $G_d$ in $\{G_d\}$ **do**
14:      Train $\mathcal{M}(G_d)$ from scratch on $\mathcal{D}_{train}$
15:      Evaluate validation loss $L_{val}(G_d)$ on $\mathcal{D}_{val}$
16: **end for**
17: Select $G^* = \arg\min_{G_d} L_{val}(G_d)$
18: **return** $G^*$

### D.2. Experimental Setup

We performed a hyperparameter sweep for $\beta$ ranging from $10^{-4}$ to $0.5$. For visual clarity in the temporal analysis, we present five representative values:

- $\beta = 0.000$ (Baseline: pure MSE)

- $\beta = 0.005$

- $\beta = 0.008$

- $\beta = 0.010$ (Selected optimal value)

- $\beta = 0.100$ (Excessive regularization)

### D.3. Regularization and Overfitting Analysis

Figure 6 illustrates the training dynamics. A comparative analysis between the training and validation losses (Panels a vs. b) reveals the fundamental role of the Wasserstein loss in improving generalization.

**Evidence of Overfitting Prevention:** The contrast between Panel (a) and Panel (b) provides strong evidence that the pure MSE baseline ($\beta = 0$) suffers from overfitting, which is effectively mitigated by the Wasserstein term:

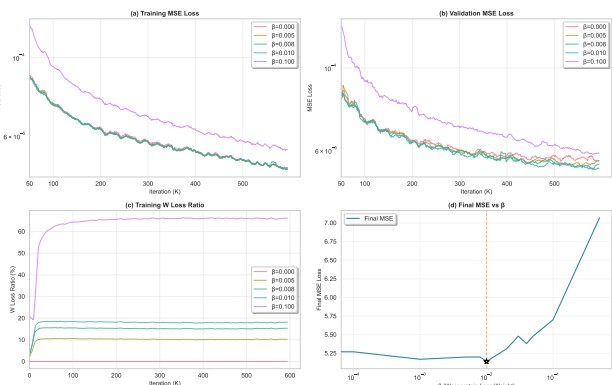

*Figure 6.* Sensitivity analysis of $\beta$. (a) Training MSE; (b) Validation MSE; (c) W Loss Ratio; (d) Final MSE vs $\beta$. Comparing (a) and (b) demonstrates that the hybrid loss prevents overfitting.

- **Training Phase (Panel a):** On the training set, the baseline model ($\beta = 0$, pink curve) performs competitively, exhibiting a loss trajectory very similar to the regularized models (e.g., $\beta = 0.01$). This indicates that the baseline model has sufficient capacity to memorize the training data.

- **Validation Gap (Panel b):** A significant divergence appears on the validation set. After approximately 400 iterations, the validation MSE for $\beta = 0$ plateaus and remains higher than the others. In contrast, models with moderate Wasserstein weights ($\beta \in [0.005, 0.01]$) continue to reduce the validation error.

This discrepancy—where training performance is similar but validation performance degrades for $\beta = 0$—confirms that the pure MSE objective leads to overfitting. The Wasserstein term acts as a crucial regularizer, constraining the model to learn robust distributional features rather than memorizing point-wise noise, thereby improving generalization on unseen data.

**Balance and Stability (Panels c & d):** While regularization is necessary, Panel (c) and (d) warn against excessive weighting.

- At $\beta = 0.100$ (purple curve), the Wasserstein term dominates the optimization (contributing $> 60\%$ of total loss in Panel c). This shifts the focus entirely to distribution matching at the expense of point-wise accuracy, causing the MSE to deteriorate significantly.

- As summarized in Panel (d), $\beta = 0.01$ achieves the global minimum, representing the optimal "sweet spot" where the regularization is strong enough to prevent overfitting but balanced enough to preserve prediction accuracy.

## D.4. Conclusion

The analysis justifies $\beta = 0.01$ as the optimal choice. The key contribution of the Wasserstein loss is identified not merely as an auxiliary objective, but as a regularizer that bridges the generalization gap, allowing the model to maintain low error rates on the validation set where the pure MSE approach fails.

