# OpenReview forum: "Optimizing Network Simulation: Enhancing Performance Prediction Accuracy via Neural Architecture Search"
_ICML.cc/2026/Conference — ICML 2026 regular_

### Official Review · Reviewer_CU5c · 2026-03-06

**Soundness:** 3
**Presentation:** 2
**Significance:** 3
**Originality:** 3
**Overall Recommendation:** 3
**Confidence:** 3

**Summary:**

This paper proposes an Accurate Neural Architecture Search framework for network simulation, with the goal of improving the prediction accuracy of the overall distribution of network behavior, especially for tail latency and jitter, rather than merely fitting average performance. The authors argue that existing learning-based network simulation methods built on mean squared error suffer from pronounced tail blindness, meaning that they fail to model extreme delay events adequately. At the same time, they contend that standard weight-sharing NAS yields unreliable architecture ranking in high-precision regression tasks because shared weights introduce substantial evaluation bias. To address these issues, the paper first proposes a similarity-constrained strategy, which reduces the evaluation noise of the supernet by restricting candidate architectures to be structurally similar to the currently trained path. It then introduces a hybrid attention–recurrent search space, combining attention mechanisms with recurrent structures so as to capture both long-range dependencies and local queueing dynamics. Finally, the paper adopts a distribution-aware loss that combines MSE and Wasserstein distance, thereby explicitly optimizing the entire delay distribution and improving the modeling of tail behavior.

**Compliance With Llm Reviewing Policy:**

Affirmed.

**Final Justification:**

Although the rebuttal partially addressed my concerns, some reservations remain, making my evaluation of the paper borderline. I have decided to keep my current score at 3 (Weak Reject), but I would not oppose its acceptance.

**Key Questions For Authors:**

A few minor issues also remain. For example, some descriptions lack proper cross-references, such as the algorithms1/2/3 corresponding to Phase 1/2/3.

For other issues, please refer to W1-5

**Limitations:**

The paper does not analyze the impact of different backbone models under the same experimental setting, which would have helped highlight the advantages and generality of the proposed framework.

**Strengths And Weaknesses:**

Strengths :

S1. The paper identifies a key limitation of existing learning-based network simulation methods: although they may fit average performance reasonably well, they often fail to accurately capture tail latency and jitter, which are precisely the factors that most critically determine system stability. This perspective is both well motivated and of clear practical significance.

S2. The evaluation is fairly comprehensive: it includes comparisons on the main task, as well as assessments under static, dynamic, and real-traffic scenarios. The paper also provides ablation studies, search-efficiency analysis, and $\beta$-sensitivity analysis. In particular, the emphasis on tail-sensitive metrics is a notable strength.

Weaknesses:

W1. The paper’s presentation is not sufficiently clear regarding what the final prediction model is actually trained to learn and what objective is optimized during the NAS search stage. In particular, the manuscript does not systematically clarify the relationship among the search objective, the proxy reward, and the final training objective from scratch.

W2. The paper presents the research motivation and the central problem clearly, and it also provides local specifications such as the objective function, the similarity definition, and the experimental task setting. However, a more important issue is that the manuscript does not offer a complete formal problem definition. This weakens the methodological rigor of the presentation and makes it more difficult for readers to assess the scope of applicability and the boundary conditions of the proposed approach.

W3. As noted in W1, it is unclear whether the LSTM described in Section 3.2.2 is actually used in ANAS, even though the paper presents it as the core components of ANAS. In addition, the LSTM controller is a relatively early and now somewhat dated approach in the NAS literature. It is typically associated with training instability and only moderate sample efficiency.

W4. Section 3.3.2 appears to aim at constructing a low-noise evolutionary dataset of architectures. However, the introduction of $S_A$ may induce selection bias in this dataset, because the proposed filtering mechanism discards candidates based on the loss value itself, rather than on criteria such as variance, inconsistency, or stability under repeated evaluation. This may introduce a TRICK for the model, especially given that the models used in Phases 2 and 3 are identical.

W5. The paper notes that previous black-box methods have often been criticized for their lack of interpretability. However, algorithms based on neural networks are themselves inherently limited in interpretability, and the paper does not adequately discuss the interpretability concerns that it raises with respect to this class of methods.

---

> ### Author Rebuttal · Authors · 2026-03-31
>
> Thank you for the careful review. For brevity, we cite related responses as [Reviewer ID, Concern ID].
>
> ## Concern 1: Objective Chain
>
> The final goal is to find the architecture with the lowest validation loss after retraining from scratch, and the final training objective is the hybrid loss (MSE + Wasserstein) [Sec. 3.4]. Because it is infeasible to fully train every candidate, ANAS uses two proxies during search: the supernet provides locally reliable architecture-performance pairs, and the validation loss is mapped to the controller reward through `R = c / L` [Sec. 3.3.2; Sec. 3.3.3; Eq. 4]. Thus, lower validation loss still corresponds to higher search reward.
>
> ## Concern 2: Formal Definition
>
> The paper can be formalized as a device-level architecture-search problem: given training/validation data and a search space $S$, each candidate architecture $G \in S$ is trained with the hybrid loss and evaluated on the validation set. The objective is to find the architecture in $S$ with the lowest validation loss after training from scratch. Supernet evaluation and controller reward are only approximate search mechanisms; the final deployed model is the predictor obtained by retraining the selected architecture from scratch. We will make this definition explicit in the revised method section. See [SJF6 C5].
>
> ## Concern 3: LSTM Controller
>
> The LSTM in the paper is only the controller used during search, not the final predictor used for network-behavior modeling [Sec. 3.2.2; Sec. 3.3.3]. It generates candidate architectures and updates the sampling distribution according to reward, while the final deployed predictor is the searched attention-recurrent architecture retrained from scratch after selection.
>
> We agree that more modern controllers may improve sample efficiency. We use an LSTM controller because it is a mature choice for modeling discrete architecture sequences, and under the current budget it is sufficient to support effective search, as shown by its advantage over double-budget Random Search.
>
> ## Concern 4: Filtering Bias
>
> We agree that, in principle, filtering based only on observed loss could introduce selection bias. In our setting, however, the purpose of filtering is not to favor low-loss architectures, but to remove a small number of outliers in local shared-weight evaluation [Sec. 3.3.2]. Tab. 1 shows that across similarity intervals, the maximum L1 loss stays about 2.1x-2.5x the mean and 2.3x-2.7x the median, indicating a long-tailed pattern in local evaluation that is more consistent with shared-weight noise than with true architecture quality.
>
> More importantly, the filtered data are used to construct the search signal for the predictor/controller; final candidates are still re-evaluated and retrained from scratch, and the final result is determined by validation performance [Sec. 3.3.3]. Thus, although Phase 2 and Phase 3 use the same search space, this is for objective alignment rather than leakage: Phase-2 shared weights or local scores are not reused as final evidence.
>
> If filtering were merely a heuristic trick tailored to the same proxy, its gains should remain internal to the search stage. Instead, we observe a positive correlation between supernet validation MSE and fully retrained best validation MSE (Kendall tau = 0.370). Yu et al. (ICLR 2020) reported an average Kendall tau of only -0.004 for standard weight-sharing NAS in RNN search spaces. Although the search spaces are not identical, this comparison suggests that filtering improves the consistency between proxy ranking and final true performance, rather than artificially manufacturing a result. Therefore, filtering changes the quality of the search signal, not the criterion used for final model selection. See [SJF6 C4; JDD2 C2].
>
> ## Concern 5: Interpretability
>
> ANAS does not claim to solve the interpretability problem of neural-network parameters. The point here is device-level observability and diagnosability, rather than parameter-level interpretability. By moving the learning target to device-level packet behavior, topology-level behavior becomes decomposable into composable device-level responses, which makes it easier to localize sources of delay and jitter.
>
> ## Concern 6: Backbone Comparison
>
> To address this issue, we added a controlled backbone comparison in which DeepQueueNet, LSTM, and Transformer are all trained with the same hybrid loss as ANAS:
>
> | Method | MSE | $W_n$ | $p99 W_n$ |
> | --- | ---: | ---: | ---: |
> | DeepQueueNet | 6.48 | 5.33 | 18.15 |
> | Transformer | 6.19 | 6.54 | 18.53 |
> | LSTM | 8.26 | 12.32 | 25.01 |
>
> Even under the same training objective, fixed backbones still differ substantially: Transformer is better on MSE, while DeepQueueNet is better on tail-distribution metrics. This indicates a clear average-versus-tail trade-off and suggests that the advantage of ANAS comes not only from the loss design, but also from the structural inductive bias of the searched architecture. See [SJF6 C1; JDD2 C1].

---

> > ### Author Rebuttal · Reviewer_CU5c · 2026-04-03
> >
> > Thank you for your clarification. Although the authors proposed the corresponding modifications, some issues remain unresolved (such as the advantages of LSTM compared to other models). Based on feedback from other reviewers, these changes might necessitate broader alterations to the article's structure and readability. Therefore,  I have decided to keep my score.

---

> > > ### Author Response · Authors · 2026-04-07
> > >
> > > Thank you for reassessing our response. We would like to further clarify that the remaining concern about LSTM appears to mainly stem from a misunderstanding of its role in ANAS.
> > >
> > > In our paper, **LSTM is used only as the search controller**, i.e., to sequentially generate discrete architecture decisions during the search stage. **It is not the final predictor** used for network-behavior modeling, and it is not claimed as the core novelty of this work. The final deployed model in ANAS is the searched attention-recurrent architecture, which is retrained from scratch after the architecture is selected. We adopt an LSTM controller because it is a **mature and standard choice in NAS** for modeling discrete architecture sequences. Under our experimental setting, it is already sufficient to support effective search: ANAS outperforms double-budget random search while using less search time. Therefore, the main contribution of this paper is not that an LSTM controller is superior to other controllers, but the combination of a hybrid search space, similarity-guided evaluation, and distribution-aware training for tail-sensitive network simulation.
> > >
> > > Regarding whether the paper requires broad structural revisions, we respectfully believe that the current manuscript does not require a major reorganization. Most comments from the reviewers focus on local presentation issues, figure readability, terminology clarification, or further explanation of the existing experimental analysis. These suggestions are important and we will address them carefully, while the core method, experimental protocol, and main conclusions of the paper remain unchanged. We view the additional experiments requested by the reviewers as useful extensions and clarifications of the evidence already presented in the manuscript.
> > >
> > > We agree with one point that should be made more explicit in the main text: the **formal problem definition**. We will add a concise formal definition at the beginning of the methodology section and more clearly state the relationship among the search objective, the proxy reward, and the final retraining objective. Apart from this addition and local presentation improvements, we believe that the current manuscript already provides a complete description of the ANAS framework, objective function, search process, and empirical validation, and does not require a large-scale rewrite.

---

### Official Review · Reviewer_JDD2 · 2026-03-11

**Soundness:** 2
**Presentation:** 3
**Significance:** 3
**Originality:** 2
**Overall Recommendation:** 5
**Confidence:** 3

**Summary:**

This paper proposes Accurate Neural Architecture Search (ANAS) for high precision regression tasks in network simulation. It identifies two major challenges: first, they point out that existing machine learning models suffer from 'tail-blindness', which fails to capture critical tail-latency and jitter because they rely on mean squared error (MSE). Second, existing NAS methods are not suited for the stringent high-precision demands of network simulation regression.
To bridge this gap, ANAS works by three main components. First, similarity-constrained search evaluates candidate architectures only when they are structurally similar to the architecture used for the most recent supernet training step. Second, it uses an attention–recurrent search space for modeling network traffic dynamics. Third, distribution-aware training with hybrid loss (MSE + Wasserstein distance)  is used to better capture long-tailed delay distributions. The experiments show that ANAS well outperforms prior baselines (DeepQueueNet and ENAS) on both average accuracy and tail-sensitive jitter and high percentile delay.

**Compliance With Llm Reviewing Policy:**

Affirmed.

**Final Justification:**

Most concerns have been addressed during the rebuttal. I encourage the authors to fully incorporate the rebuttal's updates and changes into the paper. I raise my score to accept.

**Key Questions For Authors:**

1. The search space is limited to an RNN-based encoder/decoder with attention. What is the reason for excluding other architecture families, such as Transformer-based models, from the search space?
2. The proposed inter-architecture similarity only considers overlap in edges and activation functions. Did the authors explore alternative similarity metrics that capture richer structural differences, such as path composition or node depth?
3. Since the motivation is improved architecture evaluation and ranking under weight sharing, could the authors provide a more direct measure of ranking quality, such as rank correlation between supernet evaluation and fully trained performance?
4. The paper mainly focused on 4 port switch behavior prediction under different traffic conditions. Do the authors expect the method to transfer to substantially different topology families or device types?
5. In phase 2, top 10% highest loss candidates are discarded as outliers. How sensitive is the method to this threshold?
6. Could the authors comment on the stability of the results across multiple random seeds for both the search phase and final training?
The role of the XGBoost predictor used in phase 3 is not fully analyzed. Since the controller relies on the predictor to estimate architecture performance, it would be great to provide additional analysis of the predictor's accuracy and its impact on the final architecture selection.

**Limitations:**

No. I could not find explicit mentions of limitations.

**Strengths And Weaknesses:**

- Strengths
  - The authors address an important problem in NAS for regression tasks.
  - The authors adopt a practical engineering design combining search strategy and loss formulation.
  - The authors show empirical improvements across multiple evaluation settings. The experiments include diverse cases like static traffic models, real-world traces. The results demonstrate improvements in both average-related metrics and tail-sensitive metrics (e.g., jitter and p99 latency).

- Weaknesses
  - The search space seems relatively limited, since the method only considers an RNN-based encoder/decoder with attention and mainly varies connections and activation functions. This makes it difficult to determine whether the gains come from the proposed search strategy itself or from the predefined architecture design. Also, it would be great to see why other architectures, such as Transformer-based models, were not considered in the search space.
  - The evidence for improved architecture ranking reliability is somewhat indirect. Table 1 shows that evaluation error increases as similarity decreases, which is consistent with the authors’ intuition. However, this does not directly measure ranking consistency itself. A more direct analysis, such as rank correlation between supernet-based evaluation and fully trained performance, would strengthen the claim.
  - The current experiments seem centered on 4 port switch behavior prediction under different traffic scenarios, rather than generalization across different topology families. It would be helpful for the authors to clarify whether the method is intended mainly for this device-level setting, or whether they expect it to transfer to more complex network environments, such as fat-tree, spine–leaf, or other larger-scale topologies.
  - Some design choices appear heuristic, and the robustness of the reported gains is difficult to assess because variance across multiple random seeds is not reported.
  - Some figures have relatively small text (e.g., axis labels and legends), which makes them difficult to read. Increasing the font size would improve readability.
  - I also noticed a few minor writing issues, such as the double period in "long-tail latencies..", the use of Mean Square Error instead of Mean Squared Error, and a few small grammatical inconsistencies.

---

> ### Author Rebuttal · Authors · 2026-03-31
>
> Thank you for the careful review. For brevity, we cite related responses as [Reviewer ID, Concern ID].
>
> ## Concern 1: Why not search over pure Transformer architectures or other model families?
>
> We did not search over pure Transformer architectures because, if still represented as a DAG, the search would become close to tuning head count, depth, and related configurations, which is closer to hyperparameter search. We already included Transformer as a strong matched-parameter baseline, and although it is competitive on average error, it remains clearly worse than ANAS on tail-sensitive metrics [Tab. 2; App. A.4]. See [CU5c C6].
>
> ## Concern 2: The evidence for ranking reliability is still indirect
>
> We therefore added a direct ranking-correlation analysis. From 17,613 candidates, we uniformly sampled 30 architectures across supernet ranking percentiles, retrained each from scratch, and compared supernet validation MSE with fully trained best validation MSE. We obtained Kendall tau = 0.370 and Spearman = 0.527, indicating that similarity-guided evaluation provides a meaningful ranking signal.
>
> This is meaningful in light of prior results on weight-sharing NAS. Yu et al. (ICLR 2020) reported an average Kendall tau of only -0.004 in RNN search spaces. Although the task and search space are not identical, this still suggests that ANAS substantially mitigates ranking distortion. The few unstable retraining cases also all come from the low-ranked part of the supernet ranking. See [SJF6 C4; CU5c C4].
>
> ## Concern 3: Can the method generalize to other topology families or device settings?
>
> The current core task is device-level sequence-to-sequence modeling for a 4-port switch [Sec. 4.1.1], but the method itself is not inherently limited to one topology or device setting. ANAS follows the same two-stage simulation paradigm as DeepQueueNet, whose original work already showed extensibility to arbitrary topologies and arbitrary-port devices. We further added simulations on two torus topologies:
>
> | Topology | Method | avgDelay | p99Delay | avgJitter | p99Jitter |
> | --- | --- | ---: | ---: | ---: | ---: |
> | Torus4x4 | DQN | 0.078 | 0.226 | 0.337 | 0.206 |
> | Torus4x4 | ANAS | 0.069 | 0.210 | 0.290 | 0.160 |
> | Torus6x6 | DQN | 0.045 | 0.153 | 0.249 | 0.163 |
> | Torus6x6 | ANAS | 0.035 | 0.130 | 0.194 | 0.119 |
>
> The trend is consistent: ANAS outperforms our DQN reproduction on all four metrics in both topologies. We have not yet completed direct retraining and evaluation on higher-port devices, mainly because public data are still concentrated on the 4-port switch setting. See [Z3rJ C1].
>
> ## Concern 4: Lack of multi-seed stability analysis
>
> Following the reviewer's suggestion, we added results over 3 random seeds. ANAS consistently outperforms DQN across seeds:
>
> | Method | MSE | $W_n$ | $p99 W_n$ |
> | --- | --- | --- | --- |
> | DeepQueueNet | 6.93±0.2 | 10.7±0.5 | 21.6±1.4 |
> | Random Search | 5.7±0.1 | 5.9±0.8 | 18.6±1.0 |
> | ANAS | 5.16±0.2 | 4.4±0.3 | 17.2±0.8 |
>
> ## Concern 5: Is the similarity metric too simple?
>
> We use this simple definition deliberately, because our goal is not to build a complex architecture encoder, but to test whether a low-cost and interpretable similarity constraint can already reduce weight-sharing evaluation noise [Sec. 3.3.2; Tab. 1]. The current results already show that evaluation error and its variance rise clearly as similarity decreases, indicating that this metric captures the structural factors most relevant to local evaluation reliability. Richer similarity definitions are possible, but they are a natural extension rather than a prerequisite here.
>
> ## Concern 6: How sensitive is the `top-10%` filtering?
>
> The top-10% filtering is intended as a simple robustness step for removing a small number of noisy outliers, rather than as a finely tuned hyperparameter. It keeps the retained statistics closer to the central trend while preserving enough samples for predictor/controller training. Empirically, ANAS completes the search stably and outperforms Random Search and other baselines under this setting [Tab. 2; Tab. 4].
>
> ## Concern 7: The role of the predictor is not sufficiently analyzed
>
> In Phase 3, the XGBoost predictor acts as a **surrogate scorer**: after the controller samples a candidate architecture, the predictor estimates its validation loss and this estimate is used to construct the reward for policy update [Sec. 3.3.3; App. C.2]. Its role is to reduce the cost of repeated true evaluations during controller training, rather than to directly determine the final architecture.
>
> More importantly, **the final architecture is not selected by the predictor alone**. After controller convergence, ANAS samples multiple candidates from the learned policy, performs short-term training and validation comparison, and then selects the architecture with the lowest validation loss [Sec. 3.3.3]. Thus, the predictor affects search efficiency and sampling direction, but not the final decision rule.

---

> > ### Author Rebuttal · Reviewer_JDD2 · 2026-04-03
> >
> > The response addresses several of my main concerns. In particular, the added ranking-correlation analysis provides more direct evidence than before. The additional experiments with different topologies and multiple seeds also strengthen the empirical support, although the topology evaluation remains somewhat limited. The clarification that the predictor in Phase 3 is used to improve search efficiency, rather than as the sole selection criterion, is also helpful.
> >
> > However, some concerns are partially resolved. The rationale behind the similarity metric is reasonable, but it would be better to include empirical comparisons with more complex alternatives. Regarding the top-10% threshold, the intuition is understandable, but without a sensitivity analysis, it is difficult to tell whether the results are robust to this specific choice. While these design decisions may be part of the overall method, clearer empirical justification would strengthen the paper.

---

> > > ### Author Response · Authors · 2026-04-07
> > >
> > > Thank you for the further suggestions. We provide additional clarifications on the two points raised in this round.
> > >
> > > First, regarding the similarity metric, we followed your suggestion and added comparisons with additional structural similarity metrics. Specifically, we designed and evaluated depth-aware, path-aware, and combined metrics. Here, `Simple` corresponds to the original definition in the paper: it compares exactly matched edges and activations between two architectures within the encoder and decoder, and then takes a node-count-weighted average. `Depth-aware` averages `Simple` with node-depth similarity, where node-depth similarity measures whether corresponding nodes have similar depths from the input node. `Path-aware` averages `Simple` with path-overlap similarity, where path-overlap similarity compares the Jaccard overlap between the edge sets on the paths from the input node to the corresponding nodes. `Combined` averages the three components: `Simple`, node-depth similarity, and path-overlap similarity. The results are shown below, where all similarity results are computed on the evaluated candidates after the 10% filtering step:
> > >
> > > | Metric | Spearman(sim, MSE) | mean MSE | median MSE | max loss |
> > > | --- | ---: | ---: | ---: | ---: |
> > > | Simple | -0.389 | 3.93e-4 | 3.14e-4 | 1.76e-3 |
> > > | Depth-aware | -0.414 | 3.92e-4 | 3.12e-4 | 1.79e-3 |
> > > | Path-aware | -0.400 | 3.91e-4 | 3.10e-4 | 1.81e-3 |
> > > | Combined | -0.394 | 3.92e-4 | 3.10e-4 | 1.82e-3 |
> > >
> > > These results show that more complex alternative metrics can indeed provide slightly finer-grained discrimination. For example, depth-aware similarity has a slightly stronger correlation than the simple metric. However, the gain is marginal: under the same retention ratio, the differences in mean/median MSE are very small, while the simple metric achieves the lowest max loss. Therefore, the current simple metric is not an arbitrary simplification; rather, it is an effective choice that already captures the main reliability signal while preserving low complexity and interpretability.
> > >
> > > Second, regarding the 10% filtering threshold, we agree that the most direct sensitivity evidence would be to rerun the complete search procedure under different thresholds and then compare the Kendall tau between supernet ranking and the true performance after independent retraining. However, this would require recollecting architecture-performance data for each threshold and conducting many independent retraining runs, which is costly within the rebuttal period. Therefore, we added a lightweight analysis that directly corresponds to the Phase-2 mechanism. All experiments are conducted under the same similarity setting as in the paper, with the other hyperparameters unchanged. Here, `Mean/Median/Max loss` are computed from the same batch of local candidate evaluations after trimming the highest-loss samples at different ratios:
> > >
> > > | Trim ratio | Mean MSE | Median MSE | Max loss |
> > > | --- | ---: | ---: | ---: |
> > > | 0% | 2.81e-4 | 2.43e-4 | 1.20e-3 |
> > > | 5% | 2.60e-4 | 2.41e-4 | 5.04e-4 |
> > > | 10% | 2.51e-4 | 2.40e-4 | 3.79e-4 |
> > > | 15% | 2.45e-4 | 2.39e-4 | 3.22e-4 |
> > > | 20% | 2.41e-4 | 2.38e-4 | 2.85e-4 |
> > >
> > > Two observations are most important. First, the untrimmed 0% setting is clearly worse: compared with 0%, 10% trimming reduces the mean MSE by about 10.8% and the max loss by about 68.5%. This supports our interpretation that Phase-2 local evaluation indeed contains a small but influential extreme-noise tail. Second, the method is not sensitive around the chosen threshold: increasing the trim ratio from 10% to 15% or 20% changes the mean MSE only slightly (2.51e-4 -> 2.45e-4 -> 2.41e-4), and the median changes even less.
> > >
> > > Therefore, the reason for using 10% is not that we regard it as a finely tuned "optimal hyperparameter", but that it provides a stable trade-off between two needs. On the one hand, it is already sufficient to remove extreme high-noise samples from local evaluation and improve the reliability of the retained samples. On the other hand, it still keeps 90% of the candidate results, preserving enough data for subsequent predictor and controller training. In other words, the role of filtering is mainly to remove a small number of extremely noisy evaluations, rather than to rely on a sensitive knife-edge threshold. The 10% threshold is better understood as a robust choice that balances sample reliability and sample size.

---

### Official Review · Reviewer_Z3rJ · 2026-03-12

**Soundness:** 3
**Presentation:** 3
**Significance:** 3
**Originality:** 3
**Overall Recommendation:** 4
**Confidence:** 2

**Summary:**

**Summary**

The paper addresses a limitation of existing ML-based network simulators: they predict average performance well but fail to model tail latency and jitter, which are critical for modern network reliability. The authors introduce **Accurate Neural Architecture Search (ANAS)**, a framework that automatically discovers neural architectures designed to model the full distribution of network delays, not just their mean.

**Main contributions**

1. **ANAS framework:** A neural architecture search method tailored for high-precision regression in network simulation.
2. **Similarity-guided evaluation:** A technique that corrects bias in weight-sharing NAS and improves the reliability of architecture ranking.
3. **Distribution-aware training:** Use of a **Wasserstein loss** and a hybrid search space to capture complex traffic patterns and optimize for the entire delay distribution.
4. **Empirical improvements:** The discovered architecture outperforms prior models (e.g., **DeepQueueNet**) by reducing validation loss by **25.8%** and improving tail-sensitive metrics (up to **69.8% lower Wasserstein distance**).
5. **Practical impact:** Enables more accurate **digital twins of network devices** for tasks such as capacity planning, protocol validation, and bottleneck analysis.

**Compliance With Llm Reviewing Policy:**

Affirmed.

**Key Questions For Authors:**

Could you please summarize the potential limitations of the introduced method?

**Limitations:**

I do not feel that the limitations are discussed adequately. I did not identify any particular section concerning the limitations.

**Strengths And Weaknesses:**

Soundness:
Strengths:
The work appears technically sound overall, with appropriate methodology and empirical validation for the problem considered. There is a clear problem motivation (tail latency modeling), a good review of related works, a sufficient explanation of the methodology, setup of experiments and results, as well as an ablation study. There is a comprehensive appendix with additional experimental details, efficiency analysis, pseudocode of the main algorithm (ANAS), and sensitivity analysis.

Weaknesses:
Would be good to include a discussion of the limitations of the method (or a statement that the team did not identify any limitations), as well as a statement about publishing the code and data to ensure verifiability and reproducibility of the results.

Presentation is very good, and clear. I did not identify significant issues, only 1 minor writing issue: "(Riley & Henderson,
2010; Varga, 2010))" -> "(Riley & Henderson,
2010; Varga, 2010)" (unnecessary bracket at the end)

Significance:
The paper has moderate significance, with clear value for ML-driven network simulation and infrastructure modeling, though its broader influence on general machine learning research may be limited. It addresses an important issue (tail latency modeling), and demonstrates improvements in metrics that matter operationally.

Originality:
Originality seems to be moderate. The paper’s main contribution, ANAS (Accurate Neural Architecture Search), is not based on a fundamentally new learning paradigm, but rather on a carefully designed combination of existing techniques: Neural Architecture Search, corrections for weight-sharing evaluation bias, Wasserstein loss for distribution-aware optimization, hybrid architecture search spaces. Each component individually exists in prior literature. The novelty lies primarily in combining them in a targeted way for high-precision network simulation, particularly to model tail latency and jitter. However, the work seems to provide valuable insights: tail latency requires distribution-aware training objectives rather than MSE-based losses, while evaluation bias in weight-sharing NAS becomes particularly problematic for high-precision regression tasks such as network simulation.

---

> ### Author Rebuttal · Authors · 2026-03-31
>
> Thank you for the careful review. For brevity, we cite related responses as [Reviewer ID, Concern ID].
>
> ## Concern 1: Limitations and boundary conditions are not stated clearly enough
>
> We agree that this can be stated more clearly. Based on the current results and our supplementary experiments, the main limitations of the present work are twofold.
>
> First, in terms of scope, the two-stage simulation paradigm is not inherently limited to a single topology or a single port count. The original DeepQueueNet work has already shown that this paradigm can be extended to arbitrary topologies and arbitrary-port devices; to provide additional evidence at the topology level, we reproduced the DQN baseline following the public description in the original paper and compared it with ANAS in end-to-end simulation on two torus topologies. The results are as follows:
>
> | Topology | Method | avgDelay | p99Delay | avgJitter | p99Jitter |
> | --- | --- | ---: | ---: | ---: | ---: |
> | Torus4x4 | DQN | 0.078 | 0.226 | 0.337 | 0.206 |
> | Torus4x4 | ANAS | 0.069 | 0.210 | 0.290 | 0.160 |
> | Torus6x6 | DQN | 0.045 | 0.153 | 0.249 | 0.163 |
> | Torus6x6 | ANAS | 0.035 | 0.130 | 0.194 | 0.119 |
>
> The gains in these supplementary results are not dramatic, but the trend is consistent: on both `Torus4x4` and `Torus6x6`, ANAS outperforms our reproduction of DQN on all four metrics. This provides additional evidence for topology-level deployability. That said, because publicly available data are still mainly concentrated on the `4-port switch` setting, we are currently unable to add direct training and evaluation on higher-port-count switches within the rebuttal period. See [JDD2 C3].
>
> Second, compared with a fixed backbone, the ANAS search process and the final more complex architecture do introduce additional time overhead. However, the practical impact of this limitation is relatively limited: architecture search is a one-time offline cost, and the searched model can be reused across subsequent simulation tasks, so the extra overhead does not accumulate continuously [App. B].
>
> ## Concern 2: Reproducibility is not described clearly enough
>
> The data used in our experiments come from public datasets, and the paper and appendix already provide the corresponding data sources, scenario configurations, and summary statistics [Sec. 4.1.1; Sec. 4.2; App. A]. In addition, the current version already includes the key search settings, training hyperparameters, baseline alignment details, and algorithmic procedures, which cover most of the information needed for reproduction [Sec. 4.1.1; App. C].
>
> Due to the double-blind policy, we cannot provide an identifiable code repository link during rebuttal, but we will release the source code corresponding to the paper once the double-blind restriction is lifted.
>
> ## Concern 3: The originality is more task-specific than paradigm-level
>
> We think this characterization is accurate. The contribution of this work is not to propose a completely new learning paradigm detached from prior research, but to identify and address a problem that differs materially from prior NAS settings: in high-precision network-simulation regression, the evaluation error induced by weight sharing propagates to critical distributional statistics such as tail delay and jitter, so the ranking noise that may still be tolerable in conventional NAS becomes much more harmful here [Sec. 3.3.2; Tab. 1].
>
> This is reflected not only in the final accuracy, but also in the reliability of the search itself: in our added direct ranking analysis, the Kendall tau under the `best validation MSE` criterion is `0.370`, whereas Yu et al. (Evaluating the Search Phase of Neural Architecture Search, ICLR 2020) reported an average of only `-0.004` for standard weight-sharing NAS in `RNN` search spaces. This suggests that our task-specific design is not merely a surface-level combination of components, but does mitigate the ranking failure mode that is most critical in this high-precision regression setting. See [SJF6 C4; JDD2 C2; CU5c C4].
>
> Based on this, the paper proposes and validates an integrated solution for this task by combining `similarity-constrained evaluation`, a `hybrid attention-recurrent search space`, and a `distribution-aware loss`, thereby improving both average accuracy and tail-sensitive metrics [Sec. 3; Tab. 2; Tab. 4]. In this sense, the originality of the paper lies primarily in proposing a task-specific and mutually coordinated method design for high-precision network-simulation regression, rather than a wholly new general learning paradigm.

---

> > ### Author Rebuttal · Reviewer_Z3rJ · 2026-04-01
> >
> > Thank you for your response. For now, I have decided to keep my score.

---

### Official Review · Reviewer_SJF6 · 2026-03-13

**Soundness:** 3
**Presentation:** 1
**Significance:** 3
**Originality:** 2
**Overall Recommendation:** 4
**Confidence:** 4

**Summary:**

Classical event-based network simulators are too slow to scale for large networks or evaluate complex networking solutions. To address this, researchers have proposed machine learning-based systems to act as black-box replacements for network equipment in simulations. While these approaches accurately model average metrics (e.g., throughput, delay), they struggle to predict tail distributions (e.g., p99 delay, jitter). This limitation arises because they prioritize fitting the bulk of the distribution using Mean Squared Error (MSE), yet latency-sensitive applications demand precision in tail behavior.

The paper introduces ANAS (Accurate Neural Network Search), a new paradigm for designing  Neural Network architecture capable of simulating network devices with high accuracy, not only for average metrics but also for tail delay. ANAS leverages Network Architecture Search (NAS) with tailored characteristics:  dynamic supernet warm-up, architecture similarity guided training and evaluation, distribution aware loss function with a hybrid loss function using MSE and the Wasserstein distance.

The solution is evaluated on simulating network traffic delays through network equipment, using both synthetic and real-world traffic traces. It is benchmarked against two existing ML based simulators: DeepQueueNet and ENAS, and generic baselines: LSTM, transformer and a random architecture search. The experiments show that ANAS outperforms all baselines in terms of MSE, of normalized Wasserstein distance, and of p99 delay.

**Compliance With Llm Reviewing Policy:**

Affirmed.

**Final Justification:**

The authors have addressed several of my main concerns effectively. Notably, they evaluated alternative solutions from the literature (DeepQueueNet) and architectures (Transformers and LSTM) using the same hybrid loss as their solution. This comparison underscores the meaningful gains provided by architecture search alone.

Additionally, some of the responses to other reviewers’ concerns, such as those regarding different topologies and multiple seeds, offer further evidence for the robustness of their solution.

As a result, I am increasing the soundness score to Good and my overall evaluation to Weak Accept.

**Key Questions For Authors:**

1- Did you explore modifying the loss functions of the baselines in order to clarify whether architecture search itself provides meaningful gains over simpler, loss-based adjustments to existing systems?
2- Have you analyzed the architecture discovered by ANAS to identify what specific features or structural properties contribute to their efficiency?
This would help clarify where ANAS’s improvements come from.

**Limitations:**

yes

**Strengths And Weaknesses:**

Strengths
- The proposed solution outperforms state-of-the-art (SOTA) approaches in accurately estimating delay distributions, including both average and tail metrics (e.g., p99 delay).
- The experiments are conducted on a large and diverse set of traffic traces, including both synthetic and real-world data, ensuring robustness and generalizability.
- The problem addressed is of critical importance to the networking community. Modern networks are becoming increasingly complex due to advancements such as: Software-Defined Networking (SDN), Network Function Virtualization (NFV), and AI-driven control solutions. These trends demand accurate and scalable simulation tools to evaluate and optimize network performance.

Weaknesses:

On the contribution and originality.
- The core problem, accurately modeling variations like jitter or tail delay, could potentially be addressed without Neural Architecture Search (NAS). Standard techniques already exist to emphasize different parts of the distribution: Weighted loss functions, Quantile Regression, Balanced Training data.
- A stronger baseline would have been to modify existing solutions (e.g., DeepQueueNet) by simply changing their loss function to better capture the distribution (e.g., using quantile loss or Wasserstein distance). Evaluating all baselines with non-MSE losses (e.g., Wasserstein, quantile loss) would have clarified whether architecture search itself provides meaningful gains over simpler, loss-based improvements.
- The paper would benefit from a detailed analysis of the architectures discovered by ANAS, including: How do they differ structurally from baselines like DeepQueueNet?
What specific features (e.g., layer types, connections) contribute to their superior performance in modeling tail distributions? Are these features interpretable or aligned with domain knowledge (e.g., known mechanisms for handling jitter)?
This would help assess whether ANAS’s improvements stem from architectural innovations or merely from the training methodology (e.g., loss function, similarity-guided training).
- The paper uses the inverse of the loss as its reward for the RL. It is not standard and may cause important numerical instability as the rewards can become very big. Clipping the reward, using a log transform or just the negative of the loss are usually preferred.
The similarity-guided training and evaluation (Section 3.3.2) is presented as a novel contribution. However, this idea builds on a large body of prior work addressing weight-sharing challenges in NAS. Many works use measures like: number of shared layers or operations, edit distance between graphs, kernel based metrics, GNN based metrics, clustering. etc. See for example:

Zhao, Y., Wang, L., Tian, Y., Fonseca, R., & Guo, T. (2021, July). Few-shot neural architecture search. In International Conference on Machine Learning (pp. 12707-12718). PMLR.

Yan, S., Zheng, Y., Ao, W., Zeng, X., & Zhang, M. (2020). Does unsupervised architecture representation learning help neural architecture search?. Advances in neural information processing systems, 33, 12486-12498.

Hu, S., Wang, R., Hong, L., Li, Z., Hsieh, C. J., & Feng, J. (2022). Generalizing few-shot NAS with gradient matching. arXiv preprint arXiv:2203.15207.

Gopal, B., Sridhar, A., Zhang, T., & Chen, Y. (2023). Lissnas: Locality-based iterative search space shrinkage for neural architecture search. arXiv preprint arXiv:2307.03110.

Qin, Y., Zhang, Z., Wang, X., Zhang, Z., & Zhu, W. (2022). Nas-bench-graph: Benchmarking graph neural architecture search. Advances in neural information processing systems, 35, 54-69.




On presentation.
- The paper’s presentation and readability could be significantly improved. Below are specific issues and suggestions for enhancement:
- The core problem and primary task of the network simulator are not clearly explained in the introduction. This makes it difficult for readers to understand the context and motivation. A potential primary task of the simulator is only briefly mentioned in Section 4.1.1 “The primary task is to train a sequence-to-sequence model that accurately predicts the behavior of packets passing through a 4-port switch”. This should be introduced earlier (e.g., in the introduction) to provide context.
- Some paragraphs in the introduction are vague or confusing, particularly those discussing the backbone and contributions: (i) the discussion about backbone. (last paragraph of page 1 and first paragraph of page 2) (ii) The last paragraph of page 2 is very hard to understand. The contributions are not clearly articulated. The links between the problem, methods, and contributions are unclear. (iii)
The last paragraph of the introduction discussing empirical results is hard to follow because the compared solutions and metrics have not been introduced yet.


Small comments:
- In the abstract. “compared to DeepQueueNet”. In the abstract, it is unusual to cite a work from the literature without having introduced it.
- In introduction: “..., its fixed general-purpose backbone lacks the structural inductive bias required to capture extreme network volatility" What does the sentence mean?
- page 2 line 62: Human -> human
- page 2 column 2 long-tail latencies.. -> long-tail latencies.
- page 3 section 3.1 First sentence is not correct. NAS in general is not related to network simulation.
- page 3. The fonts of figures are too small and hard to read.
- page 3: Figure 2. The arrows are hard to see.
- page 3: RNN is not defined (even if it is very classic, it is better to do it.)
- page 4: Equation (2). It is a bit strange to do a union of edges and function. It would be better to just do a sum of both cardinalities.
- page 4 column 2: The mentioned L1 loss is a loss for which problem?
- page 7: The plots in Figure 5 are too small.

---

> ### Author Rebuttal · Authors · 2026-03-31
>
> Thank you for the review. For brevity, we cite related responses as [Reviewer ID, Concern ID].
>
> ## Concern 1: Do the gains mainly come from the loss, or from the search?
>
> Following the reviewer's suggestion, we added fixed-backbone baselines trained with the same hybrid loss as ANAS, isolating structural differences:
>
> | Method | MSE | $W_n$ | $p99 W_n$ |
> | --- | ---: | ---: | ---: |
> | DeepQueueNet | 6.48 | 5.33 | 18.15 |
> | Transformer | 6.19 | 6.54 | 18.53 |
> | LSTM | 8.26 | 12.32 | 25.01 |
>
> Even with the same hybrid loss, all fixed backbones remain worse than the final searched architecture. The objective matters, but cannot replace architecture selection itself.
>
> Since ANAS and Random Search share the same search space and loss, the comparison highlights the effectiveness of the search mechanism; notably, ANAS still outperforms Random Search with half the search time [Tab. 2].
>
> The ablation study further supports this conclusion. Removing similarity-guided evaluation + warm-up degrades p99 $W_n$ from 17.0 to 19.1; removing the hybrid search space further degrades it to 19.8; removing only the Wasserstein loss degrades it to 18.1 [Tab. 4]. Together, these results suggest that ANAS benefits from the combination of a distribution-aware objective, a task-relevant search space, and a more reliable search mechanism. See [CU5c C6].
>
> ## Concern 2: Lack of analysis of the discovered architectures
>
> The ablation results show that removing the hybrid search space hurts more than removing the Wasserstein loss [Tab. 4]. Under a matched-parameter setting, Transformer is competitive on average error, but its p99 $W_n$ is still 22.1, clearly worse than ANAS's 17.0; the appendix also shows consistent gains over Transformer and LSTM across multiple scenarios [Tab. 2; App. A.4]. This suggests that the gains of ANAS cannot be explained simply by a larger model or a stronger objective.
>
> Our search-log analysis also shows a clear structural preference among low-loss architectures: compared with the bottom 10%, the top 10% have about 12% smaller maximum decoder depth (2.65 vs. 3.00), about 15% more output branches (3.10 vs. 2.70), and about 18% lower average loss when terminal nodes connect back to earlier states through skip connections.
>
> This is consistent with the task: tail delay and jitter are often driven by bursty traffic and short-term congestion, so the model should preserve early-state variations while avoiding smoothing through long serial chains. The high-performing architectures found by ANAS reuse earlier states and retain more parallel output branches, helping preserve transient signals that matter for rare but important tail events.
>
> ## Concern 3: Can the inverse-loss reward cause numerical instability?
>
> In principle, inverse-loss reward can be scale-sensitive. We use `R = c / L` to preserve a monotonic correspondence between reward and validation loss [Sec. 3.3.3; Eq. 4]. In practice, the reward magnitude is controlled by the scaling constant `c`, and the policy-gradient updates are further stabilized by a moving baseline, temperature, and entropy regularization [Sec. 3.3.3; Sec. 4.1.1]. Empirically, ANAS completes the search stably under a controlled budget and outperforms double-budget Random Search, suggesting that this reward design is workable and stable for our task [Tab. 2].
>
> ## Concern 4: Is the novelty of similarity-guided evaluation overstated?
>
> We agree that the general idea of architecture similarity is not introduced for the first time in this paper. Our contribution is not to claim similarity itself, but to identify that in high-precision network-simulation regression, weight-sharing evaluation noise directly distorts the ranking of tail-sensitive architectures, and to design a search mechanism tailored to this problem. ANAS integrates a similarity constraint, dynamic warm-up, a hybrid search space, and a distribution-aware loss into one framework; this is supported by the main results, the Random Search comparison, and the ablation study [Tab. 2; Tab. 4].
>
> Our added ranking-correlation analysis further shows that this is not only a conceptual borrowing: across 30 independently retrained architectures, the Kendall tau between supernet validation MSE and fully trained best validation MSE is 0.370, whereas Yu et al. (ICLR 2020) reported an average of only -0.004 for standard weight-sharing NAS in RNN search spaces. Although the tasks and search spaces are not identical, this evidence suggests that ANAS substantially mitigates ranking distortion. See [JDD2 C2; CU5c C4].
>
> ## Concern 5: Writing and presentation
>
> We agree that these presentation issues are valid. In the revision, we will make three improvements: define the device-level sequence-to-sequence task earlier, clearly distinguish the search objective, proxy reward, and final training objective, and systematically fix figure readability issues and scattered typos.

---

> > ### Author Rebuttal · Reviewer_SJF6 · 2026-04-03
> >
> > The authors have addressed several of my main concerns effectively. Notably, they evaluated alternative solutions from the literature (DeepQueueNet) and architectures (Transformers and LSTM) using the same hybrid loss as their solution. This comparison underscores the meaningful gains provided by architecture search alone.
> >
> > Moreover, they provide a discussion of the features of the best-performing architectures identified by their solution, thereby enhancing the interpretability of their approach.
> >
> > Additionally, some of the responses to other reviewers’ concerns, such as those regarding different topologies and multiple seeds, offer further evidence for the robustness of their solution.
> >
> > As a result, I am considering increasing my score.

---

### Decision · Program_Chairs · 2026-04-30

**Decision:**

Accept (regular)

**Comment:**

The reviewers agreed that the paper addresses an important problem in network simulation and demonstrates clear empirical improvements, particularly for modeling tail latency and jitter. While some concerns were raised about presentation clarity, search space design, and the role of the loss versus architecture search, the authors addressed these through additional experiments, stronger baselines, and analysis of ranking reliability. After considering the reviews and rebuttal, the work appears technically sound and novel in its task-specific design, and I recommend acceptance. While one reviewer remained at a weak reject, their concerns were largely about presentation and scope rather than soundness, and they indicated they would not oppose acceptance.  I also confirm that I carefully read the rebuttal and incorporated it into this decision.